# Pervasive tissue-, genetic background-, and allele-specific gene expression effects in *Drosophila melanogaster*

Amanda Glaser-Schmitt[1]*, Marion Lemoine[2], Martin Kaltenpoth[2], John Parsch[1]

**1** Division of Evolutionary Biology, Faculty of Biology, Ludwig-Maximilians-Universität München, Munich, Germany, **2** Department of Insect Symbiosis, Max-Planck-Institute for Chemical Ecology, Jena, Germany

* glaser@bio.lmu.de (AGS)

**Data Availability Statement:** The gut RNA-seq and microbiome data that support the findings of this study are publicly available from National Center for Biotechnology Information (NCBI) under the

## Abstract

The pervasiveness of gene expression variation and its contribution to phenotypic variation and evolution is well known. This gene expression variation is context dependent, with differences in regulatory architecture often associated with intrinsic and environmental factors, and is modulated by regulatory elements that can act in *cis* (linked) or in *trans* (unlinked) relative to the genes they affect. So far, little is known about how this genetic variation affects the evolution of regulatory architecture among closely related tissues during population divergence. To address this question, we analyzed gene expression in the midgut, hindgut, and Malpighian tubule as well as microbiome composition in the two gut tissues in four *Drosophila melanogaster* strains and their F1 hybrids from two divergent populations: one from the derived, European range and one from the ancestral, African range. In both the transcriptome and microbiome data, we detected extensive tissue- and genetic background-specific effects, including effects of genetic background on overall tissue specificity. Tissue-specific effects were typically stronger than genetic background-specific effects, although the two gut tissues were not more similar to each other than to the Malpighian tubules. An examination of allele specific expression revealed that, while both *cis* and *trans* effects were more tissue-specific in genes expressed differentially between populations than genes with conserved expression, *trans* effects were more tissue-specific than *cis* effects. Despite there being highly variable regulatory architecture, this observation was robust across tissues and genetic backgrounds, suggesting that the expression of *trans* variation can be spatially fine-tuned as well as or better than *cis* variation during population divergence and yielding new insights into *cis* and *trans* regulatory evolution.

## Author summary

Genetic variants regulating gene expression can act in *cis* (linked) or in *trans* (unlinked) relative to the genes they affect and are thought to be important during adaptation because they can spatially and temporally fine-tune gene expression. In this study, we used the fruit fly *Drosophila melanogaster* to compare gene expression between inbred parental

BioProject accession number PRJNA1094401. Gut non-ASE gene count and expression data are available from NCBI Gene Expression Omnibus (GEO) under the series accession number GSE263264. Malpighian tubule RNA-seq data are available under the NCBI GEO accession number GSE103645. All other relevant data are within the manuscript and its Supporting Information files.

**Funding:** This work was supported by a Deutsche Forschungsgemeinschaft (DFG, www.dfg.de) grant (number 274388701) to JP as part of the priority program "SPP 1819: Rapid evolutionary adaptation", and a DFG grant (KA2846/5-1, project number 347368302) to MK as part of FOR2682 "Seasonal temperature acclimation in Drosophila". The funder had no role in study design, data collection and analysis, decision to publish, or preparation of the manuscript.

**Competing interests:** The authors have declared that no competing interests exist.

strains and their offspring in order to characterize the basis of gene expression regulation and inheritance. We examined gene expression in three tissues (midgut, hindgut, and Malpighian tubule) and four genetic backgrounds stemming from Europe and the ancestral range in Africa. Additionally, we characterized the bacterial community composition in the two gut tissues. We detected extensive tissue- and genetic background-specific effects on gene expression and bacterial community composition, although tissue-specific effects were typically stronger than genetic background effects. Genes with *cis* and *trans* regulatory effects were more tissue-specific than genes with conserved expression, while those with *trans* effects were more tissue-specific than those with *cis* effects. These results suggest that the expression of *trans* variation can be spatially fine-tuned as well as (or better than) *cis* variation as populations diverge from one another. Our study yields novel insight into the genetic basis of gene regulatory evolution.

## Introduction

Gene expression variation is extensive at all organismal levels, including among tissues [1–2], cells [3–4], or alleles [5–6] of the same individual, and underlies much of the phenotypic variation that we see among individuals, populations, and species [7–9]. A long-standing challenge in evolutionary genetics has been to identify and characterize this variation. Indeed, elucidating the scope and architecture of gene expression variation as well as the mechanisms that shape it is an integral part of better understanding complex phenotypic traits [10–12], such as body size or disease susceptibility, and their evolution.

At the DNA sequence level, genetically heritable variants can modulate expression in two general ways: *cis*-regulatory variants, such as those within enhancers or promoters, affect the expression of linked, nearby genes, while *trans*-regulatory variants, such as those affecting transcription factors or regulatory RNAs, affect the expression of unlinked genes that can be located anywhere in the genome (reviewed in [13–14]). One way to interrogate the relative contribution of these types of regulatory variants to gene expression variation in species such as *Drosophila*, where inbred, relatively isogenic strains are available, is to compare gene expression of two parental strains or species as well as expression of their alleles in F1 hybrids [15]. Due to linkage with the allele they regulate, *cis*-regulatory variants affect only one of the two F1 hybrid alleles, leading to allele-specific expression (ASE), while *trans*-regulatory variants equally affect both alleles in the hybrid and do not lead to ASE. While *cis*-regulatory variation is thought to accumulate and become more predominant over larger evolutionary distances, i.e. between species [16–18], *trans*-regulatory variation tends to be more common among individuals within a species [5–6,19]. However, deviations from this pattern of regulatory variation have been documented in *Drosophila* [20–23] as well as other species [24–25], which underscores that there remains much to learn about the evolution of gene expression regulation, especially over short evolutionary distances.

An advantage of utilizing ASE to investigate the regulation of gene expression is that both the genetic basis of expression variation (e.g. *cis* versus *trans*) and the mode of expression inheritance (e.g., dominance versus additivity) can be assessed. Indeed, previous studies of ASE in *Drosophila* utilizing expression in F1 hybrids have found that environment [6,26], sex [27–28], genetic background [19,21,28], and body part or tissue [21–22,28] can affect regulatory architecture. However, previous studies have largely focused on single populations, long term lab strains, or comparatively closely related populations [5–6,20–22,26–28] (for an exception see [19]). Moreover, previous studies measured expression in whole animals, body parts

(e.g. heads), single tissues, and/or highly functionally diverged tissues (i.e. testes versus ovaries or heads); thus, little is known about how regulatory architecture and inheritance vary among individual tissues that are spatially and/or functionally proximate. To investigate the effect of natural genetic variation from divergent populations on regulatory architecture in multiple functionally related, interconnected tissues, we analyzed messenger RNA-sequencing (mRNA-seq) data of midgut, hindgut, and Malpighian tubule tissues in four *D. melanogaster* strains and their F1 hybrids. Two of the strains were from a population in Umeå, Sweden [29], representing the northern edge of the species' derived distribution, while the other two strains were from a population in Siavonga, Zambia, representing the species' inferred ancestral range [30]. Since their divergence from ancestral populations ~12,000 years ago [31], derived *D. melanogaster* populations have had to adapt to new habitats, and previous studies have found evidence that at least some of the expression divergence detected between derived and ancestral African populations is adaptive [32–36].

The midgut, hindgut, and Malpighian tubules, which are analogous to the mammalian small and large intestines and kidneys, respectively, physically connect to and interact with one another at the midgut-hindgut junction and are part of the *D. melanogaster* digestive tract (midgut and hindgut, together with the foregut) and excretory system (hindgut and Malpighian tubules). Both systems play important roles in the regulation of homeostasis as well as the immune response [37–38] and the investigated tissues are known to engage in interorgan communication with each other, as well as with other tissues [37–39]. The excretory system is involved in waste excretion as well as ionic- and osmoregulation [38], while the digestive tract is an important modulator of food intake, nutrient absorption, energy homeostasis, and insulin secretion that can shape physiology and behavior through its interaction with the microbiome [37,40]. To investigate the effect of natural genetic variation from divergent populations on digestive tract microbiome composition, we further performed microbiome sequencing on the same gut samples for which we performed mRNA-seq. In both the mRNA-seq and microbiome data, we found extensive tissue- and genetic background-specific effects. From the ASE data, we found that genes with both *cis* and *trans* effects were more tissue-specific than genes with no differential expression regulation, although *trans* effects were more tissue-specific than *cis* effects. Despite the context specificity that we detected for regulatory architecture across tissues and genetic backgrounds, the increased specificity of *trans* effects was consistent, suggesting that *trans*-regulatory variation can be spatially fine-tuned as well as or, potentially, better than *cis*-regulatory variation.

## Results

We performed mRNA-seq in the midgut and hindgut of two isofemale *D. melanogaster* strains from the northern limit of the derived species range in Sweden (SU26 and SU58) and two strains from the ancestral species range in Zambia (ZI418 and ZI197) as well as F1 hybrids between the Swedish and Zambian strains (SU26xZI418, SU26xZI197, SU58xZI418, and SU58xZI197). We additionally reanalyzed previously published mRNA-seq data from the Malpighian tubule [19] in a subset of these genotypes (SU26, SU58, ZI418, SU26xZI418, and SU58xZI418). We detected 7,675–8,209 genes as expressed in the individual tissues, with 6,894 genes that could be analyzed in all genotypes in all tissues. We focus on the genes that could be analyzed in all examined genotypes and tissues unless otherwise indicated. When considering gene expression variation across all samples, biological replicates clustered strongly by tissue type (Fig 1A). Within tissues, replicates mostly clustered by genotype, although in the hindgut there was some overlap between SU58, ZI418 and their F1 hybrid, as well as SU26 and one of its F1 hybrids (Fig 1).

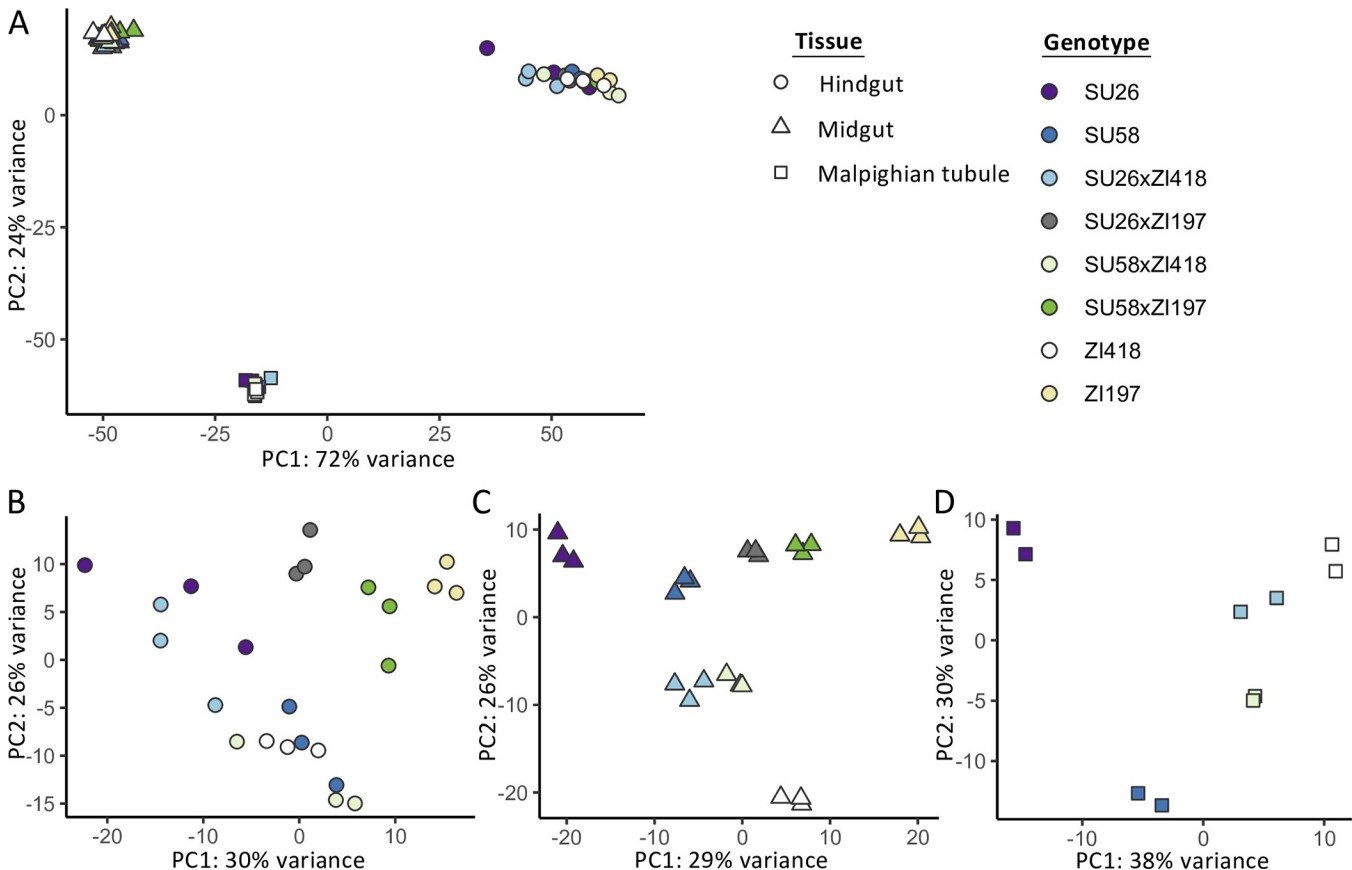

**Fig 1. Principal component analysis of gene expression profiles in A) all examined tissues and the B) hindgut, C) midgut, and D) Malpighian tubule using all genes that could be analyzed in all or each tissue(s).** The legend on the right indicates that replicates of each genotype share the same color, while shape indicates tissue.

## Differential expression among tissues and genotypes

We detected 116–2,589 (mean 961–1,398) genes as differentially expressed between genotypes within each tissue (Fig 2A). However, gene expression divergence (i.e. the cumulative differences in expression across all analyzed genes, as measured by 1 –Spearman's rho, ρ) between genotypes within each tissue was not significantly different among tissues (*t*-test; Bonferroni-corrected $P > 0.8$ for all; S1A Fig). Expression divergence tended to be lower between strains derived from the same population (i.e. Swedish strains were more similar to each other than to the Zambian strains and vice versa), although in the hindgut and Malpighian tubule, SU58 was equally or more similar to one or both Zambian strains than to the SU26 strain (Fig 2A). This pattern was not evident in the Malpighian tubule when all genes that could be analyzed in this tissue were included in the analysis (S2 Fig). When we compared expression within the same genotype among tissues, we detected 4,524–5,139 (mean 4,844) genes as differentially expressed between any two tissues (Fig 2B), 50–58% of which were differentially expressed in all pairwise tissue comparisons within a genotype and 1,619–1,880 of which were shared among at least two genotypes, with 1,045 genes differentially expressed among all tissues within all genotypes (S1 Table). Of these shared differentially expressed genes, 1,243–1,594 were consistently upregulated in the same tissue within the same genotype, with 700–924 genes consistently upregulated in the same tissue in all genotypes (S1 Table). Interestingly,

**A**

**DE genes**

| | | SU26 | SU58 | ZI197 | ZI418 | SU26 xZI197 | SU26 xZI418 | SU58 xZI197 | SU58 xZI418 |
|---|---|---|---|---|---|---|---|---|---|
| SU26 | ● | | 1093 | 1764 | 1883 | 438 | 367 | 1073 | 1240 |
| | ▲ | | 2031 | 2059 | 2589 | 1275 | 1576 | 1560 | 1797 |
| | ■ | | 1477 | – | 1914 | – | 2161 | – | 1417 |
| SU58 | ● | 0.021 | | 1451 | 1455 | 807 | 1010 | 472 | 387 |
| | ▲ | 0.021 | | 1853 | 2155 | 1133 | 1401 | 1168 | 479 |
| | ■ | 0.025 | | – | 1229 | – | 1362 | – | 226 |
| ZI197 | ● | 0.034 | 0.021 | | 1687 | 572 | 1694 | 438 | 1628 |
| | ▲ | 0.026 | 0.025 | | 2125 | 810 | 1998 | 900 | 1374 |
| | ■ | – | – | | – | – | – | – | – |
| ZI418 | ● | 0.028 | 0.019 | 0.023 | | 1351 | 692 | 1743 | 507 |
| | ▲ | 0.028 | 0.026 | 0.023 | | 2407 | 1195 | 2082 | 853 |
| | ■ | 0.028 | 0.025 | – | | – | 930 | – | 287 |
| SU26 xZI197 | ● | 0.013 | 0.014 | 0.012 | 0.020 | | 357 | 116 | 726 |
| | ▲ | 0.012 | 0.016 | 0.010 | 0.023 | | 601 | 568 | 598 |
| | ■ | – | – | – | – | | – | – | – |
| SU26 xZI418 | ● | 0.010 | 0.019 | 0.031 | 0.015 | 0.011 | | 972 | 402 |
| | ▲ | 0.014 | 0.018 | 0.023 | 0.013 | 0.009 | | 1324 | 331 |
| | ■ | 0.029 | 0.026 | – | 0.020 | – | | – | 682 |
| SU58 xZI197 | ● | 0.024 | 0.011 | 0.009 | 0.021 | 0.006 | 0.020 | | 577 |
| | ▲ | 0.019 | 0.014 | 0.010 | 0.023 | 0.008 | 0.015 | | 915 |
| | ■ | – | – | – | – | – | – | | – |
| SU58 xZI418 | ● | 0.021 | 0.009 | 0.021 | 0.010 | 0.014 | 0.014 | 0.011 | |
| | ▲ | 0.018 | 0.009 | 0.018 | 0.012 | 0.010 | 0.007 | 0.011 | |
| | ■ | 0.022 | 0.008 | – | 0.010 | – | 0.017 | – | |

*Divergence (1 − ρ)* (vertical axis label)

**B**

**DE genes**

| | MG vs HG | MG vs MT | MT vs HG |
|---|---|---|---|
| SU26 | 5091 | 4975 | 4524 |
| SU58 | 5020 | 4700 | 4533 |
| ZI418 | 5139 | 5034 | 4616 |
| ZI197 | 4988 | – | – |
| SU26xZI418 | 4946 | 4877 | 4701 |
| SU58xZI418 | 4975 | 4631 | 4599 |
| SU26xZI197 | 4968 | – | – |
| SU58xZI197 | 4881 | – | – |

**C**

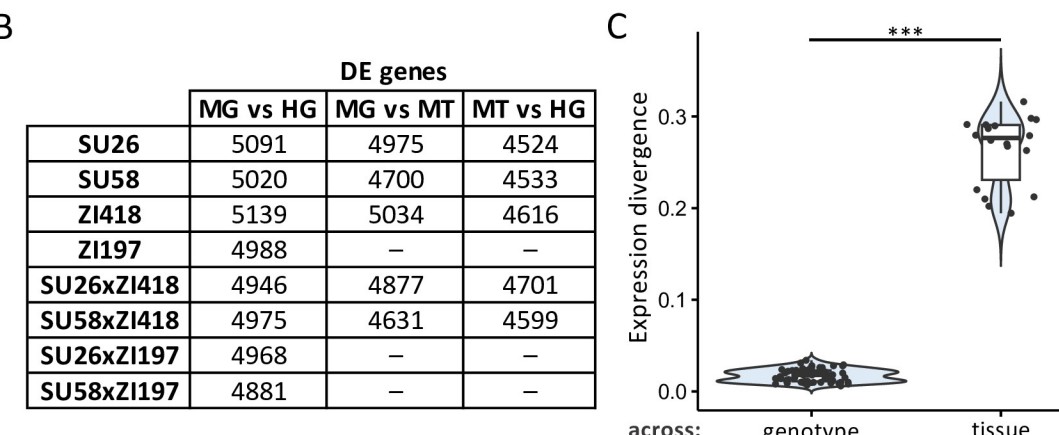

**Fig 2. Gene expression divergence among genotypes and tissues.** A) The numbers of differentially expressed (DE) genes between genotypes within the midgut (triangles), hindgut (circles), and Malpighian tubule (squares) are shown above the diagonal, while expression divergence (as measured by 1 −ρ) between genotypes is shown below the diagonal. B) The numbers of differentially expressed genes between the same genotype among midgut (MG), hindgut (HG), and Malpighian tubule (MT) tissues are shown. Dashes indicate missing data. C) Expression divergence among genotypes within the same tissue (across genotype) versus expression divergence between the same genotype among tissues (across tissue). Significance was assessed with a *t*-test. *** Bonferroni-corrected $P < 10^{-14}$.

overall gene expression divergence within the same genotype between the midgut and Malpighian tubule was significantly lower than gene expression divergence between either of these two tissues and the hindgut (*t*-test; Bonferroni-corrected $P < 5 \times 10^{-5}$ for both; S1C Fig),

suggesting that among these three tissues, expression within the same genetic background is most similar between the Malpighian tubule and the midgut. When we compared gene expression divergence among genotypes within tissues to gene expression divergence within the same genotype among tissues, gene expression divergence was higher among than within tissues (Bonferroni-corrected $P = 8.58 \times 10^{-15}$; Figs 2C and S1A). Thus, expression diverges more within a genotype among tissues than among genotypes within a tissue, suggesting that tissue is more predictive of gene expression than genotype.

## Mode of expression inheritance is highly tissue- and genetic background-specific

In order to understand how the mode of expression inheritance varies among genotypes and tissues, we categorized genes according to their expression in the two parental strains and the respective F1 hybrid into the following categories (see Methods for more details): similar, P1 dominant, P2 dominant, additive, overdominant, and underdominant, with the Swedish strains being P1 and the Zambian strains P2 (Fig 3; S2 and S3 Tables). For all backgrounds and tissues, the similar category (i.e. genes with similar expression in parents and hybrids) was the largest (Fig 3A) and showed the greatest overlap among tissues (S3 and S4 Figs). The basic expression inheritance categories (those with genes showing additive or P1 or P2 dominant expression in hybrids in comparison to parents) were the next largest categories (Figs 3A, S3, and S4), and typically similar numbers of genes were classified into these categories. However, there were some exceptions depending on category, background, and tissue (Fig 3A). For example, 1.5-fold more genes were categorized as dominant in either Swedish strain in comparison to the ZI418 strain in the midgut in comparison to the other tissues, while 3.3–4.6-fold more genes were categorized as dominant in the ZI418 strain in comparison to the SU26 strain in the midgut and Malpighian tubule in comparison to the hindgut. Similarly, 1.8–2.5-fold more genes were categorized as dominant in the ZI197 strain in comparison to either Swedish strain in the midgut than in the hindgut. Genes in these basic inheritance categories were often unique to both the tissue and category (Figs 3B, 3C, S3 and S4), with little overlap within each category across all three examined tissues (Figs 3 and S3). Unsurprisingly, in background combinations for which we only had data for two tissues, the overlap we detected within categories across tissues was higher (S4 Fig). The smallest number of genes were categorized into misexpression categories, i.e. genes showing either over- or underdominance in the hybrid in comparison to the parents (Figs 3A, 3C, S3 and S4). Genes in misexpression categories tended to be tissue-specific with little or no overlap among the examined tissues (Figs 3A, 3C, S3 and S4). Similar to what we observed for basic inheritance categories, certain combinations of genetic backgrounds and tissues showed larger numbers of misexpressed genes than others (Figs 3A, S3, and S4). For example, we detected relatively high levels of misexpression in the SU26xZI418 background in the midgut and Malpighian tubule, but not the hindgut (Fig 3A). Taken together, our results suggest that the mode of expression inheritance is both tissue- and genetic background-specific.

**Phenotypic dominance and the mode of expression inheritance.** In order to better understand potential variation in the magnitude of phenotypic dominance during expression inheritance, we calculated the degree of dominance, $h$. In order to compare the magnitude of dominance regardless of which allele was dominant, we calculated $h$ such that values between 0 and 1 or 0 and -1 represent varying degrees of additivity and dominance, with values closer to -1 representing complete dominance of the Swedish background and 1 representing complete dominance of the Zambian background, while values outside this range represent cases of overdominance of the respective background (see Methods for more details). For all genetic

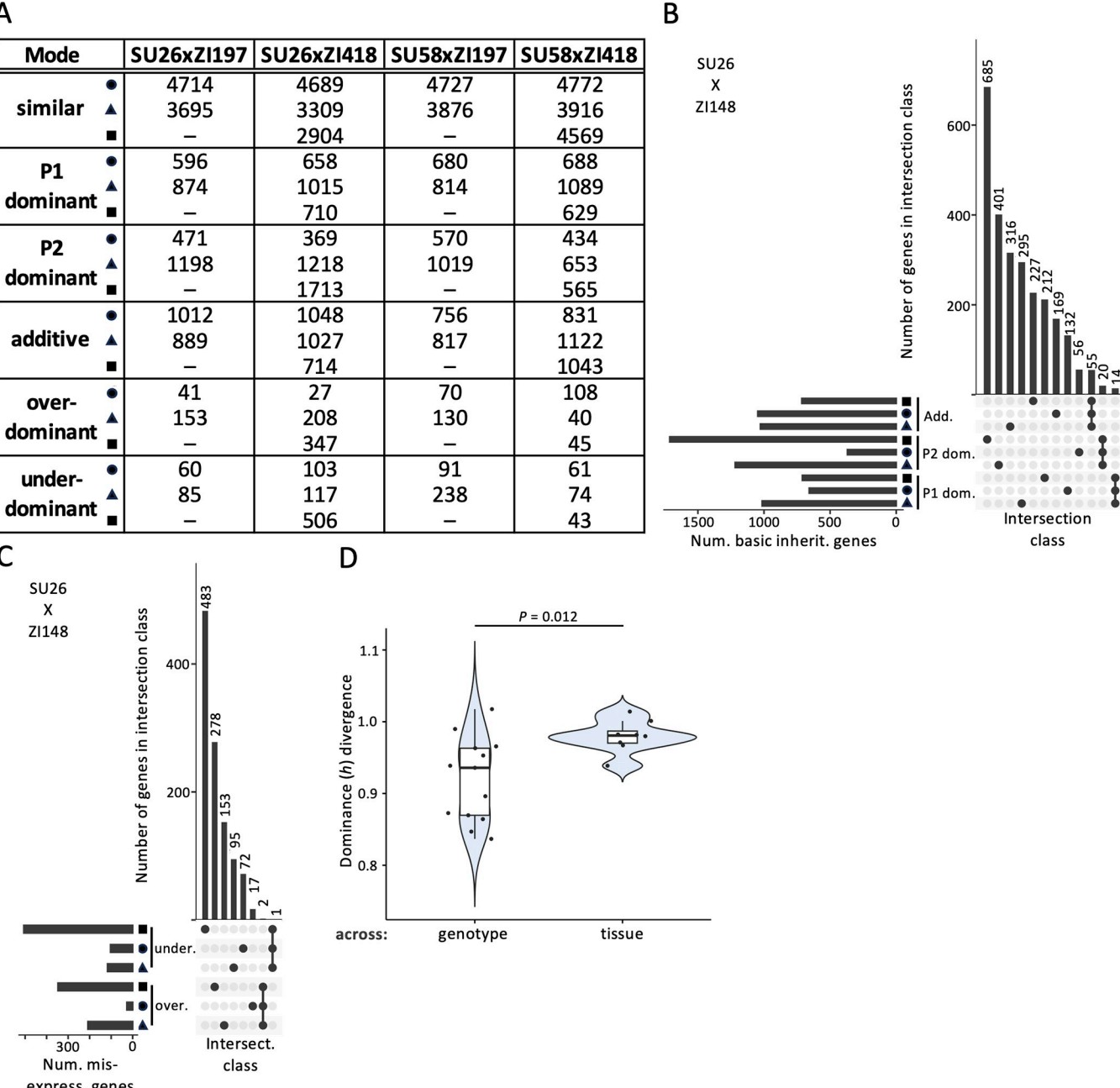

| Mode | | SU26xZI197 | SU26xZI418 | SU58xZI197 | SU58xZI418 |
|---|---|---|---|---|---|
| **similar** | ● | 4714 | 4689 | 4727 | 4772 |
| | ▲ | 3695 | 3309 | 3876 | 3916 |
| | ■ | – | 2904 | – | 4569 |
| **P1 dominant** | ● | 596 | 658 | 680 | 688 |
| | ▲ | 874 | 1015 | 814 | 1089 |
| | ■ | – | 710 | – | 629 |
| **P2 dominant** | ● | 471 | 369 | 570 | 434 |
| | ▲ | 1198 | 1218 | 1019 | 653 |
| | ■ | – | 1713 | – | 565 |
| **additive** | ● | 1012 | 1048 | 756 | 831 |
| | ▲ | 889 | 1027 | 817 | 1122 |
| | ■ | – | 714 | – | 1043 |
| **over-dominant** | ● | 41 | 27 | 70 | 108 |
| | ▲ | 153 | 208 | 130 | 40 |
| | ■ | – | 347 | – | 45 |
| **under-dominant** | ● | 60 | 103 | 91 | 61 |
| | ▲ | 85 | 117 | 238 | 74 |
| | ■ | – | 506 | – | 43 |

**Fig 3. Dominance divergence and the mode of expression inheritance in examined tissues.** A) The number of genes in each mode of expression inheritance category within the midgut (triangles), hindgut (circles), and Malpighian tubule (squares) at a 1.25-fold change cut-off. Results using alternative cut-offs or for individual tissue analyses can be found in S2 and S3 Tables (see Methods). Dashes indicate missing data. B, C) Upset plots showing unique and overlapping genes within each tissue in the SU26xZI418 background as an example. Upset plots for the other genotypes can be found in S3 and S4 Figs. Horizontal bars represent the total number (num.) of genes in a tissue and inheritance category combination. Vertical bars represent the number of genes in an intersection (intersect.) class. A filled circle underneath a vertical bar indicates that a tissue and inheritance category combination is included in an intersection class. A single filled circle represents an intersection class containing only genes unique to a single tissue and inheritance category combination, while filled circles connected by a line indicate that multiple tissue and inheritance category combinations are included in an intersection class. Genes categorized into B) basic expression inheritance (inherit.), i.e. P1 dominant (P1 dom.), P2 dominant (P2 dom.), and additive (add.) and C) misexpression (misexpress.) categories are shown. Only intersection classes comprised of either a single tissue and inheritance category combination or an inheritance category in all examined tissues are shown. Additional intersection classes and upset plots for genes categorized into the similar category are shown in S3 Fig D) Phenotypic dominance (h) divergence (as measured by 1 −ρ) among backgrounds within the same tissue (across genotype) versus dominance divergence between the same background among tissues (across tissue). Significance was assessed with a t-test. The Bonferroni-corrected P value is shown.

backgrounds and tissues, we did not detect any significant difference in the overall magnitude of phenotypic dominance between the two parental backgrounds (*t*-test, Bonferroni-corrected $P > 0.6$ for all). For the majority of tissues and genetic backgrounds, we did not detect differences in the magnitude of dominance within the same genetic background between tissues (Bonferroni-corrected $P > 0.26$ for all comparisons). We only detected a significant difference in the overall magnitude of dominance within the SU26xZI418 background between the midgut and the Malpighian tubule (Bonferroni-corrected $P = 0.015$), which may be driven by the large amount of misexpression that we detected in this background, particularly in the Malpighian tubule (Fig 3A). Overall, these results suggest that the differences we detected in the mode of expression inheritance among genetic backgrounds and tissues occur on the individual gene level rather than being driven by general, genome-wide changes in dominance. Overall dominance divergence among genetic backgrounds (i.e. the cumulative differences in dominance across all analyzed genes, as measured by $1 - \rho$) was not significantly different between the midgut and hindgut (Bonferroni-corrected $P = 0.264$, S1B Fig), but could not be compared to the Malpighian tubule for which only 2 background combinations were available. When we compared overall dominance divergence among genetic backgrounds within tissues to dominance divergence within the same genetic background among tissues, dominance divergence was significantly higher among than within tissues (Bonferroni-corrected $P = 0.012$, Fig 3D). We observed a similar pattern when we examined gene expression divergence (Fig 2C), suggesting that in general divergence is higher among tissues than among different genetic backgrounds within a tissue. However, divergence was higher for dominance than for gene expression (*t*-test, Bonferroni-corrected $P < 10^{-14}$), suggesting that phenotypic dominance of expression is much less conserved among tissues and genotypes than expression itself, although it is possible that this difference can be explained in part by differences in how each trait was measured as expression was measured in a single genotype but dominance was calculated based on three genotypes.

## Genetic basis of expression variation is highly tissue- and genetic background-specific

In order to identify genes in our dataset with any level of *cis*-regulatory divergence between the parental alleles in any genetic background and tissue, we tested for ASE in genes for which we could distinguish between the parental alleles in the hybrid (see Methods). Of the 4,305–4,592 genes we were able to analyze in all tissues of a genetic background, we detected 80–370 genes showing significant ASE (FDR <5%) depending on genetic background and tissue (Table 1), with a total of 356, 408, 460, and 256 non-redundant genes detected as having ASE

**Table 1. ASE genes.**

| Tissue[b] | SU58 vs ZI418 [a] | | | SU26 vs ZI418 [a] | | | SU58 vs ZI197 [a] | | | SU26 vs ZI197 [a] | | |
|---|---|---|---|---|---|---|---|---|---|---|---|---|
| | $DE_P$ | $DE_H$ | ASE | $DE_P$ | $DE_H$ | ASE | $DE_P$ | $DE_H$ | ASE | $DE_P$ | $DE_H$ | ASE |
| HG | 814 | 391 | 252 | 766 | 390 | 235 | 896 | 200 | 172 | 891 | 576 | 370 |
| MG | 491 | 243 | 145 | 406 | 219 | 112 | 685 | 180 | 134 | 545 | 261 | 163 |
| MT | 570 | 114 | 99 | 501 | 108 | 80 | – | – | – | – | – | – |
| All | 77 | 20 | 9 | 51 | 24 | 11 | 334 | 69 | 50 | 270 | 133 | 73 |

[a] A total of 4,035, 4,172, 4,305, and 4,592 genes could be analyzed in the SU58xZI418, SU26xZI418, SU58xZI197, and SU26xZI197 backgrounds, respectively. Results for individual tissue analyses can be found in S4 Table.

[b] Number of differentially expressed (DE) genes between the parental strains (P) and alleles within the F1 hybrid (H) as well as allele-specific genes (ASE) are shown for hindgut (HG), midgut (MG), Malpighian tubule (MT), and shared across all tissues (All). Dashes indicate missing data.

in any tissue in the SU26xZI418, SU58xZI418, SU26xZI197, and SU58xZI197 backgrounds, respectively, and a total of 958 genes in all tissues and backgrounds. Within each genetic background 55–86% of genes showing ASE in a particular tissue were unique to that tissue, while, within each tissue, 55–76% of ASE genes were unique to a single genetic background. Indeed, within each genetic background, only 9–73 genes were detected as having ASE in all examined tissues, with backgrounds in which only 2 tissues were examined sharing more ASE genes (Table 1). Thus, allele-specific expression is largely tissue- and genetic background-specific.

In order to further understand how the genetic basis of expression varies among genetic backgrounds and tissues, we classified genes in each genetic background and tissue combination into six regulatory categories: "conserved", "*cis*-only", "*trans*-only", "*cis + trans*", "*cis* x *trans*", "compensatory", and "ambiguous"[5] (see Methods for more details). The proportion of genes falling into each regulatory category was dependent upon tissue and genetic background, although, in general, when considering genes with non-ambiguous regulatory divergence in all tissues and genetic backgrounds, the largest proportion fell into the *trans*-only category which contained 2.9–30.6-fold more genes than the *cis*-only category (Fig 4A). The midgut had a higher proportion of ambiguous genes and a smaller proportion of conserved genes than the other examined tissues (Fig 4A). We detected the most *cis*-only genes in the hindgut, with 2.3–4.2 fold more genes categorized as *cis*-only in comparison to the other examined tissues (Fig 4A). In comparison to other genetic backgrounds, the SU58xZI197 background had a higher proportion of ambiguous genes and a smaller proportion of conserved genes as well as 2.2–4.7- and 2.4–10.4-fold fewer genes categorized as *cis*-only and compensatory, respectively (Fig 4A). Within each genotype, genes with non-ambiguous regulatory divergence were often unique to both the tissue and regulatory category (Figs 4, S5, and S6), with little overlap within each category across all three examined tissues (Figs 4 and S5). Similarly, within each tissue, genes with non-ambiguous regulatory divergence were often unique to a genetic background, with 35–91% of genes unique to a single genetic background within a regulatory class and tissue, while 31–87% of genes were unique to the genetic background and tissue within a regulatory class (S7 Fig). Overlap among all genotypes within a tissue was highest for genes in the *trans*-only category with 2.8–30.7-fold more shared genes categorized as *trans*-only in comparison to other non-ambiguous regulatory divergence categories (S7 Fig). Overall, our results suggest that the genetic basis of expression inheritance is both tissue- and genetic background-specific.

**Phenotypic dominance and the genetic basis of expression variation.** Previous studies have found that *cis*-regulatory variation tends to be more additive [20,27–28], while *trans* variation tends to be more dominant [27]. In order to better understand the relationship between the genetic underpinnings of expression variation and dominance, we examined phenotypic dominance (*h*; see Methods) in genes categorized as *cis*-only or *trans*-only. Overall dominance in genes categorized as *trans*-only was often biased towards one parent, with 5 out of 10 tissue and genetic background combinations significantly more dominant in one parental background than the other (Figs 4C and S8; *t*-test), and the more dominant parent dependent on the tissue and genetic background (Fig 4C). On the other hand, overall phenotypic dominance in genes categorized as *cis*-only was not significantly biased towards one parental background for any of the examined tissue and genetic background combinations (Fig 4C; *t*-test, $P > 0.5$ for all). When we compared the overall magnitude of dominance (as measured by the absolute value of dominance *h*), *trans*-only genes were only sometimes more dominant than *cis*-only genes and this was significant after multiple test correction only in the midgut of the SU26xZI418 background (Fig 4D; *t*-test, $P < 0.05$). However, this lack of significance might be due to lack of power, particularly for *cis*-only genes, of which we detected fewer (Fig 3A). When we included all genes that could be analyzed in each individual tissue and genotype in

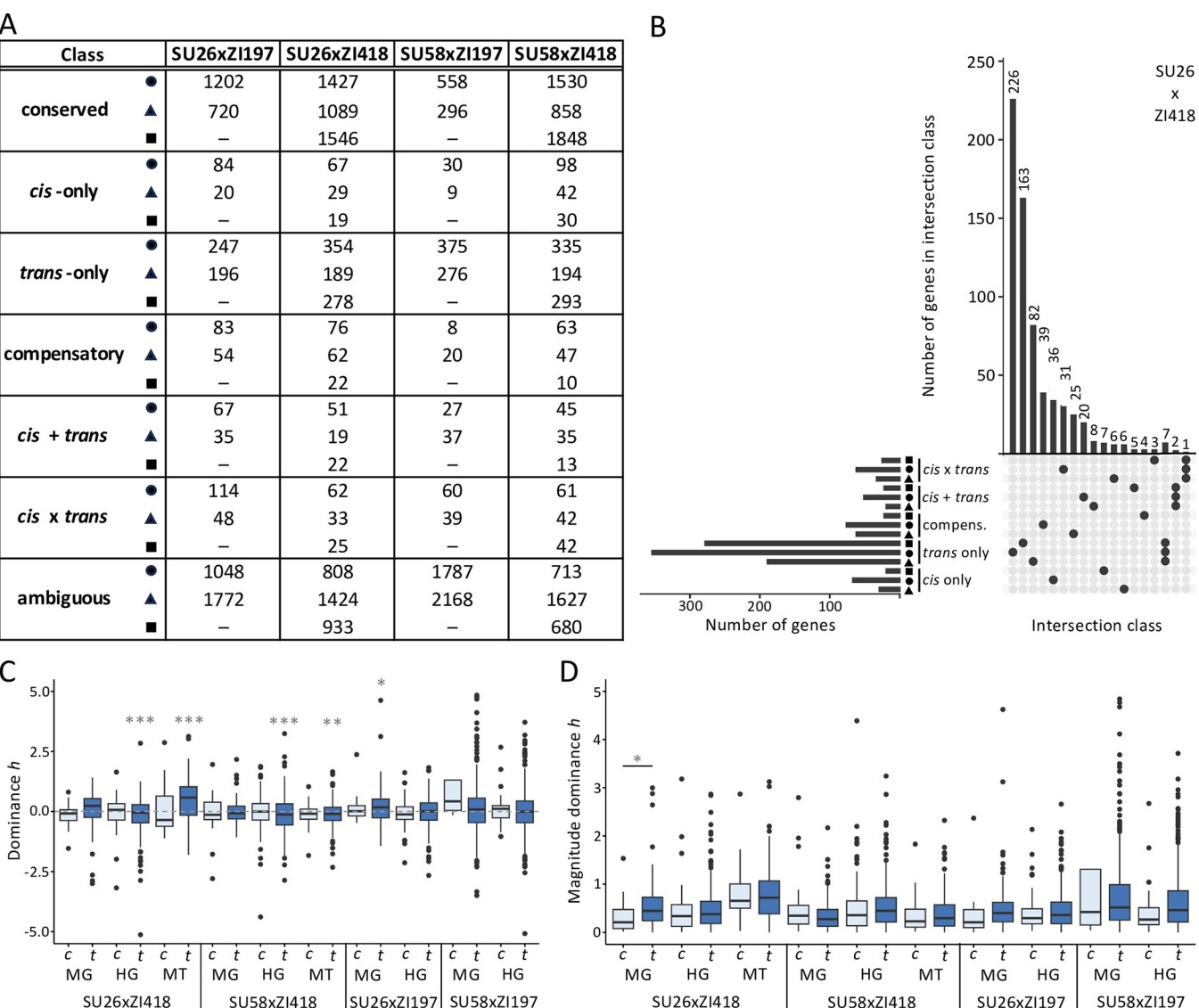

**Fig 4. Dominance and the genetic basis of expression variation.** A) The number of genes in each regulatory class within the midgut (triangles), hindgut (circles), and Malpighian tubule (squares). Results for individual tissue analyses can be found in S5 Table (see Methods). Dashes indicate missing data. B) Upset plot showing unique and overlapping genes with non-ambiguous regulatory divergence within each tissue in the SU26xZI418 background. Upset plots for other genotypes can be found in S5 and S6 Figs. Only intersection classes comprised of either a single tissue and regulatory category combination or a regulatory category in all examined tissues are shown. Additional intersection classes are shown in S5 Fig. C) Dominance and D) magnitude of dominance *h* for genes categorized as *cis*-only (*c*, light) and *trans*-only (*t*, dark) in each background and tissue. Magnitude of dominance was calculated as the absolute value of dominance *h*. Only *h* values with magnitudes of 5 or below are shown. Boxplots including more extreme *h* values can be found in S8 Fig. Significance was assessed with a *t*-test. Bonferroni-corrected *P* values are shown in grey. *** $P < 0.005$, ** $P < 0.01$, * $P < 0.05$.

the analysis, we were able to examine phenotypic dominance in 1.9–3.7-fold more *cis*-only and 1.6–2.5-fold more *trans*-only genes (S6 Table). The results, however, remained similar, with no increased detection of differences in dominance (S6 Table), which suggests that although we cannot completely rule it out, these results are unlikely to be due to a lack of statistical power. Thus, although we detected *trans*-regulatory variants as more dominant and *cis*-regulatory variants as more additive, which has been reported by previous studies [20, 27–28], we only detected this trend in a single background and tissue. Moreover, the phenotypic

dominance of *trans*- but not *cis*-regulatory variants tended to be biased toward one parental background.

**Functional classification of genes displaying ASE.** In order to understand the types of genes showing ASE in our dataset, we tested for an enrichment of gene ontology (GO) biological process and molecular function terms for genes with ASE in each background and tissue. The most commonly enriched GO terms across all backgrounds and tissues were related to oxidoreductase activity (S7 Table). Indeed, we detected at least one oxidoreductase activity term for every genetic background and tissue combination in which we detected enriched GO terms. For genes displaying ASE in all examined tissues within a genotype, we could only detect two enriched GO terms, oxidoreductase activity and response to toxic substance, in the SU26xZI197 background (S7 Table). Thus, despite ASE genes tending to be tissue- and genetic background-specific, in general ASE genes tended to be enriched for genes predicted to be involved in oxidoreductase activity.

**ASE genes are enriched for sex-biased gene expression in a context-dependent manner.** A previous study on ASE *in D. melanogaster* found differences in the relative contribution of *cis*-regulatory effects among genes with different levels of sex bias and among two tissues/body parts [28], while another using hybrids between *D. simulans* and *D. mauritiana* found that sexually dimorphic regulatory effects are often in *cis* [27]. In order to better understand the relationship between sex-biased expression and *cis*-regulatory variation in the midgut, hindgut, and Malpighian tubule, we categorized genes according to their level of sex bias using data from FlyAtlas2 [41] (see Methods). When considering genes displaying ASE (Table 1), in the Malpighian tubule we detected a significant enrichment of sex-biased genes for both genetic backgrounds ($\chi^2$ test, $P = 0.001$ for both; S8 Table), and in the hindgut we detected an enrichment of sex-biased genes in all genetic backgrounds ($P < 0.02$ for all; S8 Table) except SU26xZI418 ($P = 0.2929$; S8 Table). In the midgut, we detected a significant enrichment of sex-biased genes only in the SU26xZI197 background ($P = 0.0116$; S8 Table), despite the midgut having more sex-biased genes, particularly male-biased genes, than the other tissues (S8 Table). In the tissues and genetic backgrounds where we detected an enrichment of sex-biased ASE genes, the enrichment did not appear to be driven by bias towards one particular sex ($<1.63$-fold difference in the prevalence of male- and female-biased genes for all), with the exception of the SU26xZI197 background in the midgut, where male-biased genes were 6-fold more prevalent than female-biased genes (S8 Table). It should be noted, however, that only females were used in our experiments to measure ASE. Overall, genes displaying ASE were enriched for sex-biased gene expression, however this enrichment was dependent upon the tissue and genetic background in which they were detected.

**The effects of inversions on regulatory variation.** Chromosomal inversions have been shown to affect expression in *Drosophila* [42]. Two inversions that are at high frequency worldwide and in sub-Saharan Africa, respectively [43], were present in our study: *In(2L)t* in SU26 [19] and *In(3R)K* in ZI197 [44]. A previous study utilizing the current Malpighian tubule data found that while *In(2L)t* made a minor contribution to gene expression variation, its presence could not explain the patterns of gene expression detected in F1 hybrids with SU26 as a parent [19]. To better understand how the presence of these inversions potentially affects expression and regulatory variation, we tested for a significant over- or underrepresentation of genes differentially expressed or displaying ASE between parental strains among genes located within these inversions. Genes differentially expressed between parental strains were significantly enriched for genes located within both inversions for all comparisons, including comparisons between strains that did not contain the focal or any inversion ($\chi^2$ test, $P < 10^{-15}$ for all; S9 Table). This finding suggests that the 2L and 3R chromosome arms are enriched for differentially expressed genes, rather than the inversions themselves, which is in line with a

previous study that found that linked genetic variation within inversions is what drives differential expression rather than the structural variation itself [45]. For genes displaying ASE (Table 1), we did not find any dearth or enrichment for genes located within either inversion ($P > 0.29$; S9 Table), with the exception of the SU58xZI418 genetic background in the Malpighian tubule, which was significantly enriched for genes located on the *In(2L)t* ($P = 0.003$; S9 Table), despite having the standard chromosomal arrangement. Thus, the presence of the *In (2L)t* and *In(3R)K* inversions does not appear to explain the regulatory patterns that we detected.

## Tissue specificity varies depending on regulatory type and genetic background

For genetic background combinations for which we had transcriptome data in all three tissues (SU26xZI418 and SU58xZI418), we were able to examine the relationship between regulatory variation and tissue specificity. To do so, for every gene in each strain we calculated the tissue specificity index τ, which ranges in value from 0 to 1, with higher numbers indicating higher tissue specificity. When we compared overall τ among all genetic backgrounds, tissue specificity in ZI418 was higher than in either Swedish background as well as both F1 hybrids, although this difference was not statistically significant for SU26 (Fig 5A). On the other hand, tissue specificity in the Swedish strains was not significantly different from each other or their respective hybrid (Fig 5A). Thus, tissue specificity in F1 hybrids was more similar to the Swedish than the Zambian parent. In order to better understand how tissue specificity varies based on regulatory variation type, we performed pairwise comparisons of τ between genes with *trans*-only variation, *cis*-only variation, and conserved gene regulation. Both *cis*- and *trans*-regulated genes were significantly more tissue-specific than conserved genes for all strains in all backgrounds (Fig 5B). Interestingly, genes with *trans*-only regulatory variation were more tissue-specific than genes with *cis*-only regulatory variation, although this difference was not significant for SU58 and the F1 hybrid in the SU58xZI418 background ($P = 0.099$ and $0.076$, respectively; Fig 5B, S10 Table). Thus, *trans* effects were more tissue-specific than *cis* effects in our dataset. Unlike for the genetic basis of expression variation itself (Fig 4), we detected very few tissue-specific or tissue-by-regulatory type interaction effects on tissue specificity

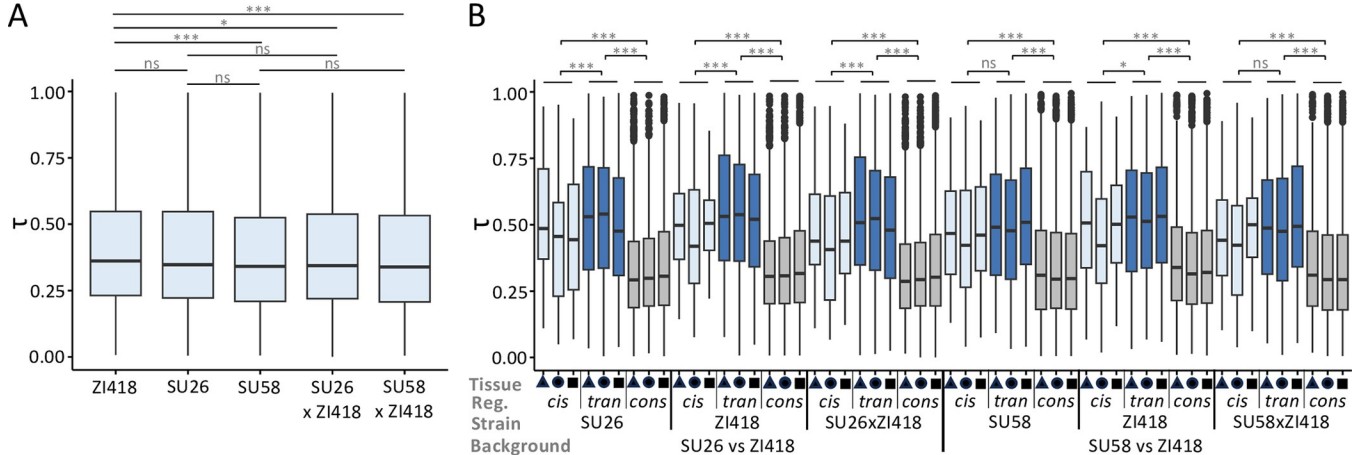

**Fig 5. Tissue specificity in the SU26xZI418 and SU58xZI418 backgrounds.** A) Overall tissue specificity as measured by τ in the examined strains. Significance was assessed with a *t*-test and Bonferroni-corrected *P* values are shown. B) Tissue specificity τ in each strain for genes categorized into *cis*-only (light), *trans*-only (tran, dark), and conserved (cons, grey) regulatory (reg.) classes for each genetic background in the midgut (triangles), hindgut (circles), and Malpighian tubule (squares). Significance was assessed with an ANOVA (S10 Table). *** $P < 0.005$, ** $P < 0.01$, * $P < 0.05$, ns not significant ($P > 0.05$).

(S10 Table). Thus, the genetic basis of regulation (i.e. *cis* versus *trans*) and, to a lesser degree, the genetic background are predictive of tissue specificity, while the tissue in which the regulatory variation was detected tends not to be. Indeed, the type of regulatory variation appears to have the largest influence on tissue specificity, as we were able to detect consistent patterns across tissues and genetic backgrounds (Fig 5B).

## Microbiome composition varies depending on tissue and genetic background

Bacterial community composition has been shown to affect host gene expression in the digestive tract depending upon host genotype [46]. In order to better understand the relationship between genetic background and microbiome composition, we performed microbiome sequencing in the midgut and hindgut for the same RNA samples for we which we performed mRNA-seq (see Methods). For all samples, *Wolbachia* was highly predominant in the bacterial community (10.36–99.37%; S9 Fig). In order to ensure its presence did not mask more subtle differences in diversity, we focus on analyses with *Wolbachia* removed (Figs 6 and 7), but results including *Wolbachia* were qualitatively similar (S9 and S10 Figs, S11–S13 Tables), and we did not detect any pattern of relative *Wolbachia* abundance among genotypes (lmer,

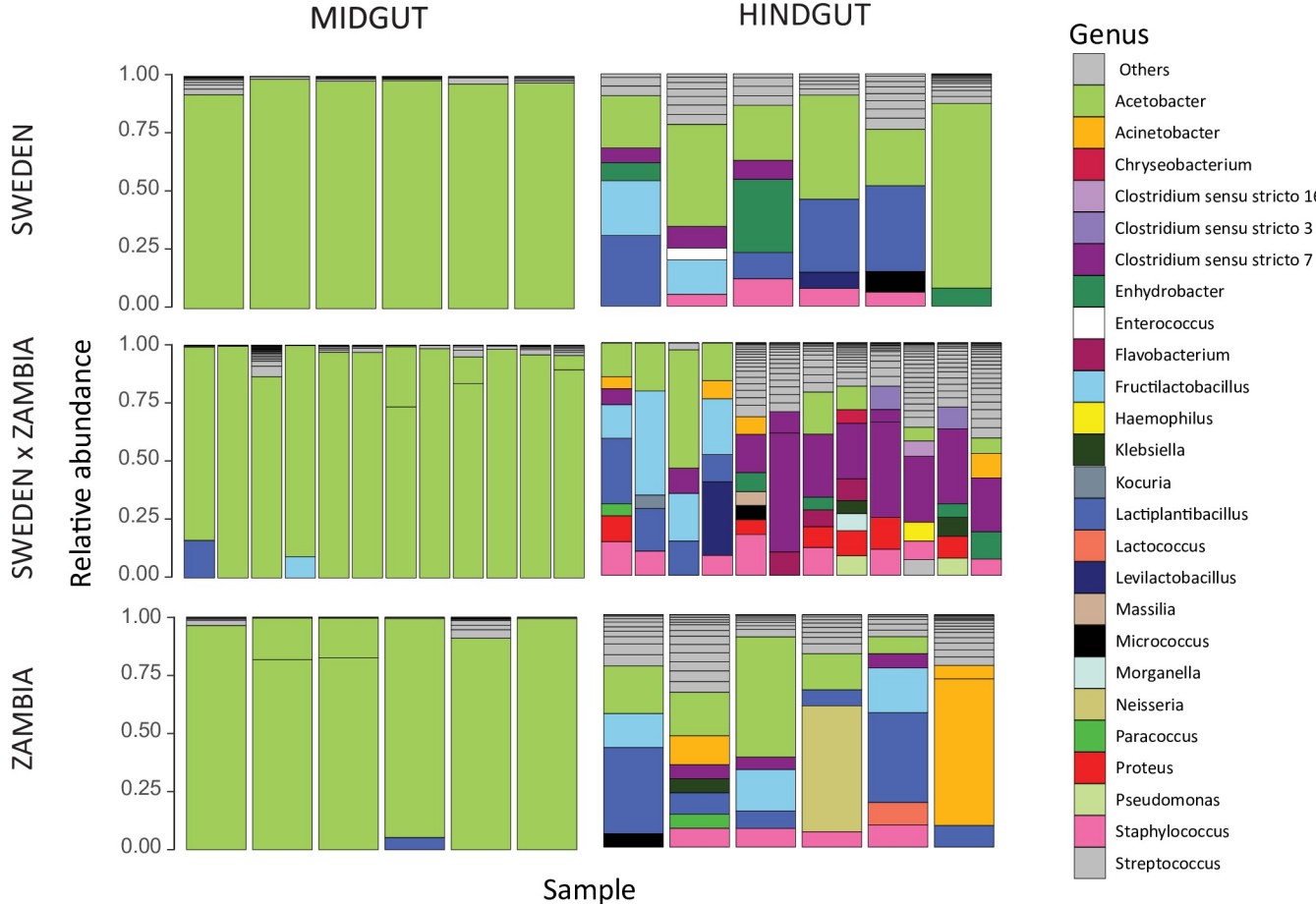

**Fig 6. Composition of the bacterial communities in the midgut and hindgut of each genotype.** Colored sections of each bar show bacterial genera (excluding *Wolbachia*) with a relative abundance above 5% in each sample. The remaining genera are compiled in the "Others" category. Bacterial community composition including *Wolbachia* can be found in S9 Fig.

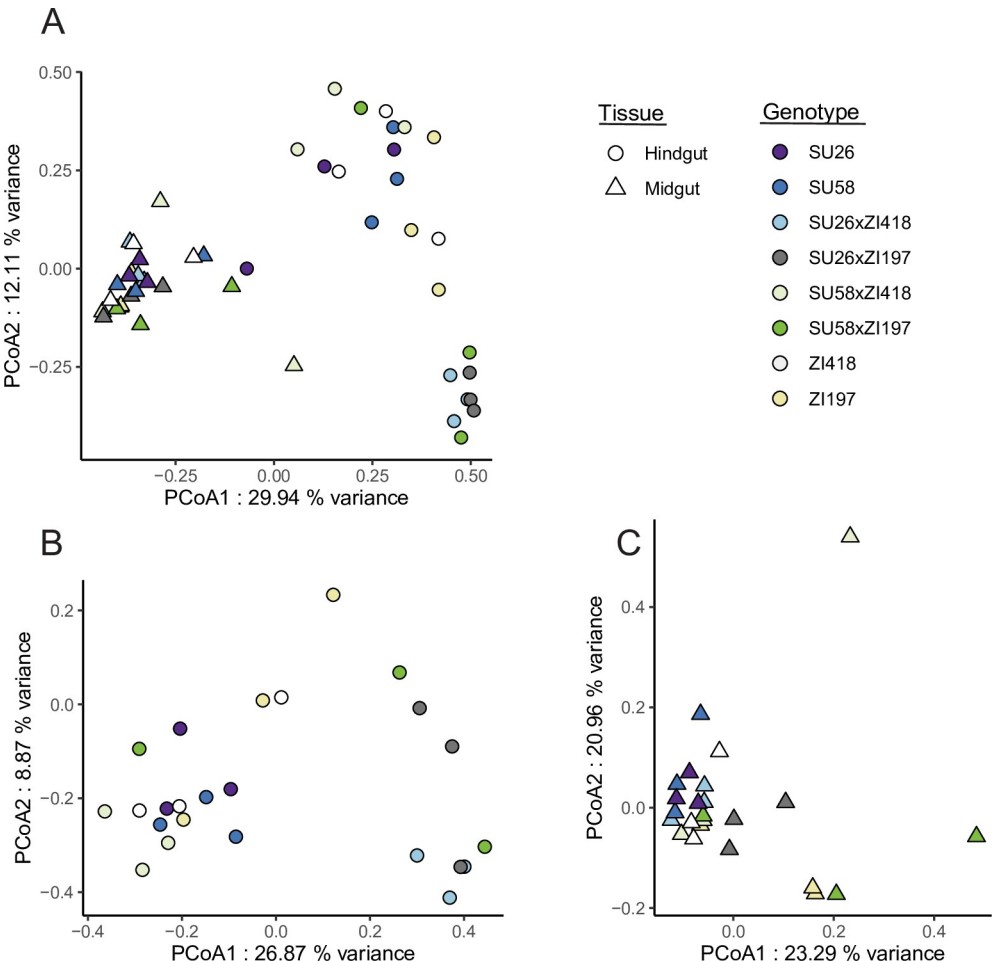

**Fig 7. Principal coordinate analysis of bacterial communities in A) both midgut and hindgut samples, B) hindgut, and C) midgut.** The legend indicates that replicates of each genotype share the same color, while shape indicates tissue. *Wolbachia* was excluded from the analysis. Results including *Wolbachia* can be found in S10 Fig.

*P* = 0.246; S14 Table). After removal of *Wolbachia*, *Acetobacter*, one of the most common *D. melanogaster* gut microbial taxa [47–48], remained predominant in the midgut (54.7–99.8%; Fig 6). In the hindgut, where the microbiome composition was more diverse, *Acetobacter* was only dominant in a subset of individuals (1.44–78.52%; Fig 6). Interestingly, we did not detect *Lactobacillus*, another of the most common gut microbial taxa [47–48]. However, because we performed amplicon sequencing using RNA rather than DNA as the starting material (see Methods), the microbiome composition we detected is representative of metabolic activity rather than presence. Thus, it may be that *Lactobacillus* was present but its metabolic activity was not high enough for us to detect.

In order to detect differences in gut bacterial community composition, we computed the Bray-Curtis index (see Methods). We detected significant differences in gut bacterial communities between the midgut and hindgut (Figs 6 and 7, S11 Table; PERMANOVA, *P* = 0.001), which is unsurprising given that these gut regions differ in their pH and associated digestive functions [37]. We also detected a significant effect of the genetic background (i.e. strain) as well as a significant interaction effect between the examined strain and gut region on the bacterial community (Fig 7, S11 Table; PERMANOVA, *P* ≤ 0.015 for both), suggesting that genetic

background affects microbiome composition, and this effect is at least partially tissue-dependent. Indeed, when we examined community composition within each tissue individually, the genetic background significantly influenced the gut bacterial community in the hindgut while it had no significant effect on the structure of the community in the midgut (Fig 7, S11 Table; PERMANOVA, $P = 0.002$ and $P = 0.89$ respectively). We also detected significant tissue and genetic background effects on bacterial alpha-diversity (S11 Fig, S12 and S13 Tables; LMM, $P = 0.017$ and $P < 0.001$), with the hindgut being more diverse and displaying stronger differentiation between parental and F1 strains (S11 Fig). Thus, the diverse bacterial community of the hindgut offered more possibility for differentiation while the midgut community was dominated by *Acetobacter* among all samples. In contrast, gene expression in the hindgut was less differentiated among genetic backgrounds but more differentiated from the other tissues while the midgut showed the opposite pattern (Figs 1 and S1C), suggesting that the expression and regulatory variation we detected in these tissues is unlikely to be driven by bacterial community composition. Thus overall, tissue type (i.e. gut region) had the largest impact on microbial community composition and diversity, with genetic background also affecting microbiome variation to a lesser degree, especially within the hindgut; however, these genetic background effects do not appear to be related to host gene expression variation.

## Discussion

Using transcriptome data from parental and F1 hybrid *D. melanogaster* strains from an ancestral and a derived population in the midgut, hindgut and Malpighian tubule, we found that both the genetic basis of expression variation (i.e. *cis* versus *trans*) and the mode of expression inheritance (i.e. dominant versus additive) were highly tissue- and genetic background-specific (Figs 3 and 4, Table 1). Previous studies using F1 hybrids in *Drosophila* have found that genetic background [19,21,28] and body part or tissue [21–22,28] can have large effects on regulatory architecture; however, to our knowledge, this is the first study examining highly spatially and functionally proximate tissues that not only communicate with each other but also functionally and physically interact. Thus, our results demonstrate that even functionally related, interconnected tissues can show highly divergent regulatory architecture among tissues and genetic backgrounds. Indeed, overall gene expression was most similar between the Malpighian tubule and the midgut (Fig 1), despite these tissues being part of the excretory and digestive system, respectively, while the two gut tissues are part of the same alimentary canal. Thus, our results suggest that the level of functional and physical interconnectivity between tissues may not necessarily be predictive of similarity in gene expression or regulatory architecture. Consistent with this interpretation, we detected similar fold-size differences in the proportion of *cis*-regulated genes in the hindgut versus the Malpighian tubule or midgut (Fig 4A) as has previously been reported in the testes versus the head or ovaries [28]. However, we should note that the tissues we examined in this study, and the midgut in particular, are known to be regionalized [37–38,40] with an estimated 22, 4, and 5 distinct cell types currently described in the midgut, hindgut, and Malpighian tubules, respectively [49–50]; therefore, it is possible that we may have missed some of the more subtle differences in gene regulation that occur among individual regions or cell types.

It has long been thought that regulatory changes and particularly *cis*-regulatory changes are important during adaptation as they can fine-tune gene expression both temporally and spatially [51–52]. Indeed, we found that genes with *trans* and *cis* effects were more tissue-specific than genes with no differential expression regulation (Fig 5), suggesting that regulatory changes between diverged populations are often tissue-specific, which is likely driven by spatial fine-tuning of gene expression. Interestingly and somewhat surprisingly, we found that *trans*

effects were more tissue-specific than *cis*-effects and this finding was consistent across tissues and genetic backgrounds (Fig 5B). Thus, our results reveal that *trans*-regulatory changes can be as or, potentially, even more tissue-specific than *cis*-regulatory changes that occur as populations diverge. In contrast to our findings, a recent study on ASE in two mouse tissues found that tissue-specific genes were largely *cis*-regulated during population divergence [25]. Indeed, *cis*-regulatory changes have long been thought to be more common during adaptation due to lower pleiotropy [53]. One of the disadvantages of our methods is that we were unable to assign regulatory effects to their underlying genetic variants and do not know the location or identity of the genetic variants underlying the detected tissue-specific *trans* effects. Thus, it is possible that the tissue-specific *trans* effects we detected are driven by *cis*-regulatory changes in the transcription factors or other regulators driving these *trans* effects.

Previous studies found that *cis*-regulatory variation tends to be more additive [20,27,28] than *trans* variation, which tends to be more dominant [27]. However, we found little evidence for this pattern in our dataset (Fig 4D). The discrepancy between the current study and previous ones may be due to differences in methods or the examined genetic background and/or body parts/tissues, suggesting that differences in additivity and dominance between *cis* and *trans* variation may be context-specific. Interestingly, we found that the phenotypic dominance of *trans*- but not *cis*-regulatory variation tended to be biased toward one parental background, with the direction of the bias variable among tissues and genetic backgrounds (Fig 4C). The context-dependent nature of this finding suggests that this bias may be driven by one or several *trans* factors affecting the expression of multiple genes in individual tissues and genetic backgrounds, which underscores the importance of taking genetic background and tissue into account when attempting to identify general patterns and trends in gene expression and its regulation. When we examined divergence in gene expression and phenotypic dominance (i.e. the cumulative differences in each trait across all analyzed genes), we found that divergence was higher among than within tissues (Figs 2C and 3D), suggesting that although both are pervasive, tissue-specific effects outnumber or are larger than genetic background-specific effects, and these effects may be magnified when considering the phenotypic dominance of gene expression, as our findings suggest that it is much less conserved than expression itself.

A previous study in *D. melanogaster* larvae found that overall developmental (i.e. temporal) gene expression specificity increased during adaptation in a derived population [36]. In contrast, in our dataset the ancestral ZI418 genetic background showed the highest tissue (i.e. spatial) gene expression specificity (Fig 5A); however, it is possible that overall changes in gene expression specificity driven by adaptation are only detectable at the population rather than the individual level. Because we identified regulatory variation between an ancestral and a derived *D. melanogaster* population that had to adapt to new habitats, some, although not necessarily all, of the regulatory variation we identified may be adaptive. Indeed, one recent study examining ASE between warm- and cold-adapted mouse strains found signs of selection on ASE genes in the cold-adapted mice [25]. Genes we identified as showing ASE included several for which adaptive *cis*-regulatory divergence has previously been documented, such as *MtnA* [54], *Cyp6g1* [55], *Cyp6a20* [19], and *Cyp12a4* [19]. We also detected an enrichment of oxidoreductase activity and response to toxic substance among ASE genes (S7 Table), suggesting any genes with adaptive *cis*-regulatory variation that we detected may be related to these processes. Indeed, the detected selection on *Cyp6g1* expression is thought to have been driven by resistance to the pesticide DDT [55], while selection on *MtnA* is thought to be driven by increased oxidative stress resistance [34,54]. Indeed, the digestive system's direct interaction with the external environment [37] and the excretory system's role in detoxification and waste excretion [38] suggest that many of the ASE genes we identified may be candidates for adaptation.

Similar to our findings for the genetic basis of expression variation, the mode of expression inheritance, and phenotypic dominance (Figs 3–5), we detected significant effects of tissue and genetic background on bacterial community composition in our microbiome analysis, although the detected genetic background effects did not appear to explain host gene expression variation (Figs 6, 7, and S11). The endosymbiont *Wolbachia*, which is known to affect microbiome composition but is not present in the gut lumen [56], was predominant in all of our samples (S9 Fig) but we did not detect the very common *Lactobacillus* (Fig 6), which suggests that physical abundance within the gut may not be predictive of metabolic activity levels and some bacterial community members may be more or less active than predicted by their physical abundance. Bacterial community composition was highly divergent between the two gut tissues, and the effect of genetic background appeared to be driven by higher diversity in the hindgut, which also showed more differentiation among strains (Figs 6, 7 and S11). Because all flies were reared in the same lab environment, a large portion of the detected bacterial community was likely acquired in the lab. Rearing environment greatly influences bacterial community composition, with communities of lab-reared strains less diverse than their natural-reared counterparts [57]. Thus, it is difficult to draw conclusions about how the genetic background effects we detected might influence bacterial community composition in nature, although genetic differentiation among natural *D. melanogaster* populations is known to shape bacterial community structure [58].

Overall, our findings yield insight into the evolution of regulatory architecture, the effects of regulatory variation on tissue specificity, the effects of genetic background on expression and microbiome variation, as well as the importance of accounting for context-specificity in evolutionary studies.

## Materials and methods

### *D. melanogaster* samples and sequencing

All *D. melanogaster* strains were reared on cornmeal-molasses medium under standard lab conditions (21˚C, 14 hour light: 10 hour dark cycle). mRNA-seq and microbiome sequencing were performed for four isofemale strains, two from Umeå, Sweden (SU26 and SU58) [29] and two from Siavonga, Zambia (ZI418 and ZI197) [30] as well as F1 hybrids between the Swedish and Zambian parental lines (SU58xZI418, SU58xZI197, SU26xZI418, SU26xZI197). The SU58 and ZI418 strains have the standard arrangement for all known chromosomal inversion polymorphisms [19, 44], while SU26 and ZI197 have the standard arrangement with the exception of *In(2L)t*, which was present in SU26 [19] and *In(3R)K*, which was present in ZI197 [44]. To determine if inversion status affected our findings, we tested for a significant over- or under-representation of genes differentially expressed or displaying ASE between parental strains among genes located within the *In(2L)t* or *In(3R)K* inversion using a $\chi^2$ test. Reciprocal F1 hybrids were generated in both directions (i.e. parental genotypes were switched) by crossing 2–3 virgin females of one line with 4–5 males of the other line. Crosses were carried out in 8–13 replicate vials and parental strains were similarly reared (2–3 females and 3–5 males per vial with 8–12 replicate vials) in order to control for rearing density among genotypes.

Midguts (from below the cardia to the midgut/hindgut junction, 20 per biological replicate) and hindguts (from the midgut/hindgut junction to the anus, 60 per biological replicate) were dissected from 6-day-old females in cold 1X PBS and stored in RNA/DNA shield (Zymo Research Europe; Freiburg, Germany) at -80˚C until RNA extraction. For F1 hybrids, half of the tissues were dissected from each of two reciprocal crosses in order to avoid potential parent-of-origin effects, although such effects are expected to be absent or very rare in *D. melanogaster* [6,59]. RNA was extracted from three biological replicates per genotype and tissue type

(48 samples in total) with the RNeasy Mini kit (Qiagen; Hilden, Germany) as directed by the manufacturer. mRNA-seq and microbiome sequencing were performed using the same RNA extractions. Poly-A selection, fragmentation, reverse transcription, library construction, and high- throughput sequencing was performed by Novogene (Hong Kong) using the Illumina HiSeq 2500 platform (Illumina; San Diego, CA) with 150-bp paired reads. Malpighian tubule 125-bp paired read data for SU58, SU26, ZI418 and F1 hybrids (SU58xZI418, SU26xZI418), which was composed of 2 biological replicates per genotype (10 in total; 58 libraries in total across all tissues) was downloaded from Gene Expression Omnibus (accession number GSE103645).

## Microbiome sequencing and analysis

Reverse transcription was carried out to generate complementary DNA (cDNA) which was used for amplicon sequencing targeting the V4 region of the 16S rRNA bacterial gene. First, template RNA was cleaned of potential residual genomic DNA with the PerfeCta DNase I (Quantabio; Beverly, MA) following the manufacturer's instructions. Reverse transcription was performed using the FIREScript RT cDNA Synthesis (Solis BioDyne; Tartu, Estonia) with specific bacterial primers, 515F (5'-GTGYCAGCMGCCGCGGTAA-3') and 806R (5'- GGAC TACNVGGGTWTCTAAT-3'), also following the manufacturer's instructions. The V4 region of the 16S rRNA gene was sequenced from the resulting cDNA on an Illumina Miseq platform using the 515F and 806R primer pair. Using the R package DADA2 (version 1.26.0) [60], Amplicon Sequence Variants (ASVs) were inferred after trimming (length of 240nt for forward reads and 180nt for reverse reads). Dereplication and chimera removal were performed using default parameters of DADA2. Each ASV was assigned taxonomically using the Silva classifier (version 138.1) [61]. ASVs assigned to the Eukaryotic and Archeal kingdoms were removed. Given that the gut bacterial community was highly dominated by one ASV assigned to the genus *Wolbachia* (S9 Fig), a known intracellular symbiont of *Drosophila melanogaster*, we chose to remove it for further statistical analysis, revealing the underlying diversity in the gut bacterial community.

All statistical analyses were performed in R-4.2.2 and each graph was generated with the *ggplot2* package [62]. The composition of the bacterial gut community was analyzed using the *Phyloseq* package [63]. ASVs not present in more than 6.25% (3 replicates/48 samples = 0.00625) of the samples were removed for visualization purposes but kept in the data for the remaining analyses. Differences in beta-diversity were tested with permutational multivariate analysis of variance (*vegan* package version 2.6–4) [64] on a Bray-Curtis dissimilarities matrix and Principal coordinate analyses (PCoA) was performed for visualization using the *vegan* package. Differences in bacterial alpha diversity (species richness, Shannon index, Simpson index and inverse Simpson index generated with *vegan* package) were tested with linear mixed models (lmer, *lme4* package) [65] and pairwise comparisons were tested following the Tukey method (*emmeans* package) [66]. The RNA extraction batch had no significant effect on differences in alpha and beta-diversity of the bacterial community.

## mRNA-seq analyses

Reference genomes for each parental strain were constructed using published genome sequence assemblies of SU26 and SU58 [29], and ZI197 and ZI418 [44, 67] as described in [19]. Briefly, if a nucleotide sequence difference on the major chromosome arms (X, 2R, 2L, 3R, 3L) occurred between a parental strain and the *D. melanogaster* reference genome (release 6) [68], the parental nucleotide variant was included in the new reference transcriptome. If the parental sequence contained an uncalled base ("N"), the reference sequence was used. All

transcribed regions (including rRNAs, non-coding RNAs, and mRNAs) were then extracted from each parental reference genome using FlyBase annotation version 6.29 [68]. For each parental strain library, mRNA-seq reads were mapped to the corresponding parental reference genome. In order to prevent mapping bias for genes with greater sequence similarity to one of the parental reference genomes, reads for F1 hybrids were mapped to the combined parental reference genomes.

Reads were mapped to the reference transcriptomes using NextGenMap [69] in paired-end mode. Read pairs matching more than one transcript of a gene were randomly assigned to one of the transcripts of that gene. For downstream analyses, we analyzed the sum of read counts across all of a gene's transcripts (across all annotated exons), i.e. on the individual gene-level. To identify genes with poor mapping quality, for each parental transcriptome, we simulated mRNA-seq data with 200 reads per transcript and either 125 bp or 150 bp reads, then mapped the reads back to the corresponding transcriptome. Genes for which more than 5% of reads mapped incorrectly were removed from the analyses of the corresponding read length (125 bp for Malpighian tubule and 150 bp for midgut and hindgut). Library size ranged from 34.7 to 55.0 million paired end reads, 97.0–98.6% of which could be mapped (S15 Table).

ASE and mode of expression inheritance analyses were performed within individual tissues as well as for all tissues together. Analyses for individual tissues were qualitatively similar to our analyses including all tissues; therefore, we focus in the main text on analyses including all tissues (S1 and S2 Data) and have included individual tissue analyses as Supplementary material (S3–S5 Data, S3–S6 Tables). To standardize statistical power across all libraries included in the analysis, we held the total number of mapped reads constant by setting the maximum number of mapped reads per sample to that of the library with the fewest mapped reads and randomly subsampling reads (without replacement) until the total number of mapped reads for each sample equaled the maximum. The number of reads we subsampled for each dataset were as follows: 34,009,757 in midgut, 31,611,417 in hindgut, and 30,820,759 in Malpighian tubule as well as for analyses including all tissues. We identified differentially expressed genes using a negative binomial test as implemented in DESeq2 [70]. Gene expression divergence between two strains or tissues was calculated as 1 –Spearman's ρ between the mean normalized gene counts of the two. Significant differences in divergence were assessed with a *t*-test. To be considered as expressed in our dataset, we required that a gene have a minimum of 15 reads in each sample, which resulted in 7,684, 8,209, 7,675, and 6,894 genes in the midgut, hindgut, Malpighian tubule, and all tissues, respectively, that could be used in analyses.

## Calculation of tissue specificity and phenotypic dominance *h*

We calculated normalized gene counts for each sample using DESeq2 [70] for genes expressed in all tissues. We then used the normalized gene counts to calculate the tissue-specificity index tau, τ, [71] for each genotype for which we had data from all three examined tissues and were able to examine tissue specificity for 3,338 genes expressed in all genetic backgrounds and tissues. The degree of phenotypic dominance (*h*) was calculated for each set of parental strains and their respective F1 hybrid (4 genetic background combinations in total) as:

$$h = \frac{2X_{F1} - X_{ZI} - X_{SU}}{X_{ZI} - X_{SU}}, \tag{1}$$

where $X_{ZI}$, $X_{SU}$, and $X_{F1}$ represent the mean normalized gene count across all replicates for the Zambian parental strain, the Swedish parental strain, and the F1 hybrid, respectively [72] in each set of background combinations. This equation for phenotypic dominance yields values between -1 (complete dominance of the Swedish background) and 1 (complete dominance of

the Zambian background), which allows for a simple and intuitive comparison of the magnitude of dominance between the two backgrounds but differs slightly from how phenotypic dominance is calculated by other methods (for example, see [73]). Divergence in the degree of phenotypic dominance ($h$) between two genetic backgrounds or tissues was calculated as 1 – Spearman's ρ between the two. Significant differences in divergence were assessed with a $t$-test.

## Inference of the mode of expression inheritance

To infer the mode of expression inheritance in F1 hybrids, we compared F1 hybrid expression to parental expression and classified genes into six categories: "similar," "P1 dominant", "P2 dominant", "additive," "overdominant," and "underdominant" [5]. To do so, we compared the fold-change difference in expression as calculated by DESeq2 [70] for each gene between genotypes to a fold-change cutoff threshold. Genes where all expression differences were below the cutoff were classified as "similar", while genes for which the expression difference was greater than the cutoff between the hybrid and only one parent were classified as dominant for that parent. Genes were categorized as additive if the expression differences between the hybrid and both parents was above the cutoff and the hybrid expression was between the expression of the two parental strains, or if the difference in expression between the two parents was above the cutoff and hybrid expression was between the two parental strains. Genes were categorized as overdominant if the expression difference between the hybrid and both parents was above the cutoff and hybrid expression was greater than that of both parents. A gene was categorized as underdominant if the expression difference between the hybrid and both parents was above the cutoff and hybrid expression was lower than that of both parents. We employed three fold-change cutoffs (1.25, 1.5, and 2) as well as a negative binomial test [70] and a 5% FDR cutoff, for which we also included an ambiguous category for genes that did not fit into the other categories. In the main text, we focus on the 1.25-fold cutoff as i) the relative proportion of genes falling into each of the non-similar categories was qualitatively similar for all cutoffs (S2 Table), ii) a fold-change cutoff (rather than a statistical test) should avoid bias in detecting differential expression between alleles/genes with higher expression, as the power of statistical tests increases with increasing read counts, iii) the 1.25-fold cutoff has been employed in several previous studies with the justification that most of the significant expression differences detected between samples tend to be of this magnitude [5,20,74], and iv) previous work using the Malpighian tubule data we use here empirically determined it to be a reasonable cutoff for this analysis [19]. The results for the other cut-offs and individual tissues are provided in S2 and S3 Tables.

## ASE analysis

In order to detect expression differences between the two alleles in each F1 hybrid, we compiled lists of diagnostic SNPs that could be used to distinguish between transcripts for each pairwise combination of parental alleles in each examined tissue (S16 Table). To do so, we first compared the two parental reference genome sequences over all transcribed regions annotated in the *D. melanogaster* reference genome (version 6.29) [68] to compile an initial list of diagnostic SNPs. In order to exclude sites with potential residual heterozygosity or sequencing errors, for each tissue, we required that all SNPs inferred from the genome sequences be confirmed in the parental mRNA-seq data with a coverage of $\geq$ 20 reads and the expected variant in $\geq$ 95% of the mapped reads of each parent. Next, we called new SNPs from the parental mRNA-seq data if a site was not polymorphic (or contained an N) in the parental genome sequences, but had $\geq$ 20 mapped reads in each parent with $\geq$ 95% having the same base in one

parent but a different base the other parent. The total number of high-confidence diagnostic SNPs meeting these criteria was 66,030–89,497, 74,388–105,634, and 59,294–60,668 for midgut, hindgut, and Malpighian tubule, covering 6,937–7,474, 7,399–8,166, and 6,394–6,412 genes, respectively.

To assess ASE in the F1 hybrids, we used the mapping data described in the mRNA-seq analyses section above, but used only reads containing at least one diagnostic SNP (i.e. reads that could be assigned to a parental allele). As described above, counts were summed over all transcripts of a gene and the ASE analysis was carried out on a per gene basis. To standardize statistical power between genetic background combinations and/or tissues while maximizing the number of reads that could be included in our analysis, the maximum number of diagnostic reads per sample (i.e. the maximum number of reads for 2 alleles) was set to that of the F1 hybrid with the fewest diagnostic reads. For all other samples, reads were randomly subsampled (without replacement) until the total number of diagnostic reads equaled the maximum for F1 hybrids or half of the maximum for parents. The maximum number of diagnostic reads was set to 15,446,286, 11,058,789, 12,477,714, and 11,058,789 for midgut, hindgut, Malpighian tubule, and all tissues, respectively. We tested for differences in allelic expression using a negative binomial test as implemented in DESeq2 [70], using only genes with a minimum of 15 diagnostic reads for each allele replicate, resulting in a total of 5,060–5,590, 5,650–6,141, 5,097–5,133, and 4,035–4,592 genes depending on genetic background combination that could be analyzed in the midgut, hindgut, Malpighian tubule, and all tissues, respectively, of which 4,228, 4,800, 4,397, and 2,845 genes could be analyzed in all genetic background combinations. In the main text, we focus on the 2,845 genes that could be directly compared across all genetic background combinations and tissues, although results for individual tissues and genetic background combinations were qualitatively similar (S5 Table).

### Inference of the genetic basis of expression variation

We determined the genetic basis of expression variation for each gene using the outcome of three statistical tests: a negative binomial test for differential expression between the two parental strains, a negative binomial test for ASE in the $F_1$ hybrid, and a Cochran–Mantel–Haenszel (CMH) test of the ratio of expression between the two parents and the ratio between the two alleles in the hybrid. For all tests, P-values were adjusted for multiple testing [75] and an FDR cutoff of 5% was used to define significant differences. We employed the same subsampling procedure as described in the ASE analysis section above in order to balance statistical power between parents and hybrids. We classified genes into regulatory classes [5] as follows: "conserved" genes showed no significant difference in any test; "all *cis*" genes showed significant ASE in hybrids and significant DE between parents, but the CMH test was not significant; "all *trans*" genes showed significant DE between the parents and a significant CMH test, but no ASE; "compensatory" genes had no DE between parents, but showed significant ASE in hybrids and a significant CMH test; "*cis + trans*" genes were significant result for all three tests with the expression difference between the parents greater than the difference between the two alleles in the hybrid; "*cis × trans*" genes also had three significant tests, but the expression difference between the parents was less than the difference between the two alleles in the hybrid; and "ambiguous" genes were significant for only one test.

### Gene set enrichment analysis

We used InterMine [76] to search for an enrichment of gene ontology (GO) biological process and molecular function terms for genes displaying ASE in each genetic background and tissue as well as in all tissues.

## Sex-biased gene analysis

In order to calculate sex bias for each gene in each examined tissue, we downloaded male (M) and female (F) FPKM (Fragments Per Kilobase of transcript per Million mapped reads) values from FlyAtlas2 [41] and calculated sex bias as $log2(FPKM_M/FPKM_F)$. Based on this log fold-change (LFC), we categorized genes as either sex-biased (LFC ≥ 0.5 or LFC ≤ -0.5) or unbiased (LFC ≤ 0.5 or LFC ≥ -0.5). We tested for a significant over- or underrepresentation of sex-biased genes among genes displaying ASE using a $\chi^2$ test with a Benjamini-Hochberg multiple test correction. To better understand how these sex-biased genes were distributed among different levels of sex bias, we further categorized genes as strongly female-biased (FS; LFC ≤ -1.5), moderately female-biased (FB; -0.5 ≤ LFC ≤ -1.5), unbiased (UB; -0.5 > LFC > 0.5), moderately male-biased (MB; 0.5 ≥ LFC ≥ 1.5), or strongly male-biased (MS; LFC ≥ 1.5; S8 Table).

## Supporting information

**S1 Fig. Expression and dominance (*h*) divergence within and among tissues.** A) Expression and B) dominance (*h*) divergence among genotypes within the midgut (MG), hindgut (HG), and Malpighian tubule (MT) versus divergence between the same genotype among tissues (across). C) Expression and D) dominance (*h*) divergence within the same genotype among tissues. A–C) Significance was assessed with a *t*-test. D) Significance was not assessed due to the low number of comparisons. Bonferroni-corrected *P* values are shown. * $P < 0.05$, ** $P < 5 \times 10^{-5}$, *** $P < 10^{-14}$, ns not significant, nt not tested.
(PDF)

**S2 Fig. Differential expression and divergence within tissues.** The total number of differentially expressed (DE) genes between genotypes within the A) hindgut (HG), B) midgut (MG), and C) Malpighian tubule (MT) are shown above the diagonal, while expression divergence (as measured by ρ subtracted from one) between genotypes is shown below the diagonal. Analysis was performed in each tissue individually. The numbers of genes that could be included in the analysis for each tissue were 8,209 in the hindgut, 7,684 in the midgut, and 7,675 in the Malpighian tubule.
(PDF)

**S3 Fig. Mode of expression Inheritance in SU26xZI418 and SU58xZI418 backgrounds.** Upset plots showing unique and overlapping genes within the hindgut (circles), midgut (triangles), and Malpighian tubule (squares) in the A,C,E) SU26xZI418 or B,D,F) SU58xZI418 backgrounds. Horizontal bars represent the total number (num.) of genes in a tissue and inheritance category combination. Vertical bars represent the number of genes in an intersection class. A filled circle underneath a vertical bar indicates that a tissue and inheritance category combination is included in an intersection class. A single filled circle represents an intersection class containing only genes unique to a single tissue and inheritance category combination. Filled circles connected by a line indicate that multiple tissue and inheritance category combinations are included in an intersection class. Genes categorized into A,B) basic expression inheritance (inherit.), i.e. P1 dominant (P1 dom.), P2 dominant (P2 dom.), and additive (add.), C,D) misexpression (misexpress.), and E,F) similar categories are shown.
(PDF)

**S4 Fig. Mode of expression inheritance in SU26xZI197 and SU58xZI197 backgrounds.** Upset plots showing unique and overlapping genes within the hindgut (circles) and midgut (triangles) in the A,B,E) SU26xZI197 or C,D,F) SU58xZI197 backgrounds. Horizontal bars

represent the total number (num.) of genes in a tissue and inheritance category combination. Vertical bars represent the number of genes in an intersection class. A filled circle underneath a vertical bar indicates that a tissue and inheritance category combination is included in an intersection class. A single filled circle represents an intersection class containing only genes unique to a single tissue and inheritance category combination. Filled circles connected by a line indicate that multiple tissue and inheritance category combinations are included in an intersection class. Genes categorized into A,C) similar, B,D) basic expression inheritance (inherit.), i.e. P1 dominant (P1 dom.), P2 dominant (P2 dom.), and additive (add.), and E,F) misexpression (mis-express.) categories are shown.
(PDF)

**S5 Fig. Genetic basis of expression inheritance in SU26xZI418 and SU58xZI418 backgrounds.** Upset plots showing unique and overlapping genes with non-ambiguous regulatory divergence in the hindgut (circles), midgut (triangles), and Malpighian tubule (squares) in the A) SU26xZI418 or B) SU58xZI418 backgrounds. Horizontal bars represent the total number of genes in a tissue and regulatory category combination. Vertical bars represent the number of genes in an intersection class. A filled circle underneath a vertical bar indicates that a tissue and inheritance category combination is included in an intersection class. A single filled circle represents an intersection class containing only genes unique to a single tissue and regulatory category combination. Filled circles connected by a line indicate that multiple tissue and regulatory category combinations are included in an intersection class.
(PDF)

**S6 Fig. Genetic basis of expression inheritance in SU26xZI197 and SU58xZI197 backgrounds.** Upset plots showing unique and overlapping genes with non-ambiguous regulatory divergence in the hindgut (circles) and midgut (triangles) in the A) SU26xZI197 or B) SU58xZI197 backgrounds. Horizontal bars represent the total number (num.) of genes in a tissue and regulatory category combination. Vertical bars represent the number of genes in an intersection class. A filled circle underneath a vertical bar indicates that a tissue and inheritance category combination is included in an intersection class. A single filled circle represents an intersection class containing only genes unique to a single tissue and regulatory category combination. Filled circles connected by a line indicate that multiple tissue and regulatory category combinations are included in an intersection class.
(PDF)

**S7 Fig. Genetic basis of expression inheritance across examined tissues and backgrounds.** Shown are A,B) unique and C,D) overlapping genes in each regulatory category. Shown are A) the number of genes unique to each tissue within each regulatory category and genetic background, B) the number of genes unique to each genetic background and tissue within each regulatory category, C) the number of genes in each regulatory category detected in all examined tissues for each genetic background, and D) the number of genes in each regulatory category detected in all genetic backgrounds for each tissue. Asterisks (*) indicate comparisons using only C) two tissues or D) two genetic backgrounds.
(PDF)

**S8 Fig. All dominance in *cis*-only versus *trans*-only genes.** A) Dominance and B) magnitude of dominance $h$ for genes categorized as *cis*-only (*c*, light) and *trans*-only (*t*, dark) in each background and tissue. Significance was assessed with a *t*-test. Bonferroni-corrected $P$ values are shown in grey. *** $P < 0.005$, ** $P < 0.01$, * $P < 0.05$, ms $P$ marginally significant after multiple test correction ($P < 0.1$), ns $P$ not significant after multiple test correction.
(PDF)

**S9 Fig. Composition of the bacterial communities in the midgut and hindgut of each genotype, including *Wolbachia* ASVs.** Colored sections of each bar show bacterial genera with a relative abundance superior to 5% in each sample. The remaining genera are compiled in the "Others" category.
(PDF)

**S10 Fig. Principal coordinate analysis of bacterial communities in A) both midgut and hindgut samples, B) hindgut, and C) midgut, including *Wolbachia* ASVs.** The legend indicates that replicates of each genotype share the same color, while shape indicates tissue.
(PDF)

**S11 Fig. Shannon diversity index of the bacterial community in the midgut and hindgut excluding (A) or including (B) *Wolbachia* ASVs.** * indicates significant differences of the Shannon index between groups (lmer, $P < 0.05$).
(PDF)

**S1 Table. Shared differentially expressed genes among tissues and genotypes.** The numbers of overlapping differentially expressed (DE) genes within or among genotypes and/or tissues for all DE genes (All) or genes upregulated in the midgut ($MG_{up}$), hindgut ($HG_{up}$), and Malpighian tubule ($MT_{up}$) tissues are shown. Only genotypes for which data was available in all examined tissues are shown.
(XLSX)

**S2 Table. Mode of expression inheritance in combined tissue analysis.** Numbers of genes in each mode of expression inheritance category within the hindgut, midgut, and Malpighian tubule at a 1.25-, 1.5-, and 2-fold change or 5% FDR cut-off (see Methods) are shown. 6,894 genes could be included in the analysis. The ambiguous category is only necessary for the 5% FDR cutoff and comprises genes which could not be assigned into other categories.
(XLSX)

**S3 Table. Mode of expression inheritance in individual tissue analysis.** Numbers of genes in each mode of expression inheritance category within the hindgut, midgut, and Malpighian tubule at a 1.25-, 1.5-, and 2-fold change or 5% FDR cut-off (see Methods) are shown. 7,684, 8,209, and 7,675 genes could be included in the analysis in the midgut, hindgut and the Malpighian tubule, respectively. The ambiguous category is only necessary for the 5% FDR cutoff and comprises genes which could not be assigned into other categories.
(XLSX)

**S4 Table. ASE genes identified in individual tissue analyses.** Number of differentially expressed (DE) genes between the parental strains (P) and alleles within the F1 hybrid (H) as well as allele specific genes (ASE) are shown for hindgut (HG), midgut (MG), Malpighian tubule (MT), and shared across all tissues (All). Dashes indicate missing data.
(XLSX)

**S5 Table. The genetic basis of expression inheritance in individual tissue analyses.** Numbers of genes in each regulatory category within the hindgut, midgut, and Malpighian tubule are shown. 4,228, 4,800, and 4,397 genes could be included in the analysis in the midgut, hindgut and the Malpighian tubule, respectively.
(XLSX)

**S6 Table. Phenotypic dominance in all *cis* and all *trans groups* including all genes that could be analyzed in each individual tissue and genotype.** The mean and mean of the

absolute value of phenotypic dominance (*h*) are shown. Significance was assessed using a *t*-test. Significant *P*-values are in bold and values non-significant after multiple test correction are shown in grey.
(XLSX)

**S7 Table. Enriched GO terms in genes with ASE.** Enriched molecular function and biological process GO terms for genes showing ASE in the Malpighian tubule (MT), midgut (MG), hindgut (HG), and all examined tissues. Number (num) of contributing terms and Holm-Bonferroni-adjusted *P*-values are shown. Genetic background and tissue combinations with no enriched GO terms are not shown.
(XLSX)

**S8 Table. Sex bias in genes displaying ASE.** The number of genes categorized as strongly female-biased (FS; LFC $\leq$ -1.5), moderately female-biased (FB; -0.5 $\leq$ LFC $\leq$ -1.5), unbiased (UB; -0.5 > LFC > 0.5), moderately male-biased (MB; 0.5 $\geq$ LFC $\geq$ 1.5), strongly male-biased (MS; LFC $\geq$ 1.5), and generally sex-biased (SB; LFC $\geq$ 0.5 or LFC $\leq$ -0.5) are shown. Significant deviations from the expected number of sex-biased genes (SB EXP) were assessed with a $\chi^2$ test. *P*- and Benjamini-Hochberg-corrected *P*-values are shown.
(XLSX)

**S9 Table. The effect of *In(2L)t* and *In(3R)K* inversion status.** The number of genes differentially expressed (DE) or displaying allele specific expression (ASE) between parental strains along with the number of genes located within an inversion are shown. The strain containing an inversion is shown in parentheses. Significant deviations from the expected number of DE or ASE genes were assessed with a $\chi^2$ test.
(XLSX)

**S10 Table. ANOVA results for pairwise comparisons between genes categorized as *cis*-only, *trans*-only, and conserved.** Results for ANOVAs with τ of the indicated strain as the response variable and regulatory (reg.) variant type, tissue, and the interaction between them as factors.
(XLSX)

**S11 Table. PERMANOVA results on comparison of Bray-Curtis distances of the bacterial community including or excluding *Wolbachia* ASVs.**
(XLSX)

**S12 Table. Summary statistics of the LMER ANOVA comparing Shannon indices of the bacterial community including or excluding *Wolbachia* ASVs.** The mixed models included the genotype and tissue as fixed factors and the batch and sample ID as random factors. The interaction term in between the two fixed effect did not have a significant effect on the Shannon indices and was subsequently removed from the models. LMER: Shannon index ~ genotype + tissue + 1| batch + 1| sample ID.
(XLSX)

**S13 Table. Pairwise comparisons of the Shannon index for the bacterial communities including or excluding *Wolbachia* ASVs.** The comparisons were performed following the Tukey method and the *P*-values were adjusted via the Benjamini-Hochberg (BH) method.
(XLSX)

**S14 Table. Results of lmer comparing the relative abundance of *Wolbachia* in the bacterial communities.**
(XLSX)

**S15 Table. Library size and mapping efficiency for all mRNA-seq libraries.** To prevent mapping bias, all libraries were simultaneously mapped to at least one pair of parental strains (P1 and P2). The number of mapped, paired mapped, unmapped, and discarded reads as well as total reads, read pairs, and proportion (prop) of mapped reads are shown.
(XLSX)

**S16 Table. Number of diagnostic SNPs in each tissue and comparison.** Shown are the number of diagnostic SNPs determined from the respective reference genome (Ref) and with the mRNA-seq data included as well as the number of genes covered by the diagnostic SNPs.
(XLSX)

**S1 Data. ASE gene counts and expression in analyses including all tissues.**
(XLSX)

**S2 Data. Phenotypic dominance, sex-biased gene expression, and τ in analyses including all tissues.**
(XLSX)

**S3 Data. ASE gene counts and expression, and phenotypic dominance in midgut analyses.**
(XLSX)

**S4 Data. ASE gene counts and expression, and phenotypic dominance in hindgut analyses.**
(XLSX)

**S5 Data. Gene counts, expression and phenotypic dominance in Malpighian tubule analyses.**
(XLSX)

**S6 Data. ASV and taxonomy tables of the bacterial communities in the midgut and hindgut.**
(XLSX)

## Acknowledgments

We thank Hilde Lainer for excellent technical assistance in the lab as well as Dr. Grit Kunert (Department of Biochemistry, Max-Planck-Institute for Chemical Ecology) for advice on statistical models. We also thank the LMU Evolutionary Biology department for helpful suggestions and discussions.

## Author Contributions

**Conceptualization:** Amanda Glaser-Schmitt, John Parsch.

**Formal analysis:** Amanda Glaser-Schmitt, Marion Lemoine.

**Funding acquisition:** Martin Kaltenpoth, John Parsch.

**Investigation:** Amanda Glaser-Schmitt, Marion Lemoine.

**Methodology:** Amanda Glaser-Schmitt, Martin Kaltenpoth, John Parsch.

**Visualization:** Amanda Glaser-Schmitt, Marion Lemoine.

**Writing – original draft:** Amanda Glaser-Schmitt.

**Writing – review & editing:** Amanda Glaser-Schmitt, Marion Lemoine, Martin Kaltenpoth, John Parsch.

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
