## [Decision Letter · Decision Letter 0]

13 Jun 2024

Dear Dr. Glaser-Schmitt, 

Thank you very much for submitting your Research Article entitled 'Pervasive tissue-, genetic background-, and allele-specific gene expression effects in *Drosophila melanogaster*' to PLOS Genetics.

The manuscript was fully evaluated at the editorial level and by independent peer reviewers. The reviewers appreciated the attention to an important topic but identified some concerns that we ask you address in a revised manuscript.

We therefore ask you to modify the manuscript according to the review recommendations. Your revisions should address the specific points made by each reviewer.

Yours sincerely,

Trudy F.C. Mackay

Consulting Editor - PLoS Genetics

PLOS Genetics

Scott Williams

Section Editor

PLOS Genetics

Reviewer's Responses to Questions

**Comments to the Authors:**

Reviewer #1: The authors examine gene expression in three related tissues in individuals from two geographically separated populations plus F1 individuals to study both tissue and allele specific effects on gene expression. The authors find a large number of differentially expressed genes between tissues and backgrounds and that tissue specific effects were larger than background effects. This finding provides insight into the role of cis and trans effects in differential gene expression. The main takeaway is that there is a lot of variability between tissues and background in terms of genes that show ASE and gene regulation. The regulatory variation vs tau comparison is quite interesting.

The methods used in the paper are straightforward and have been used in many other similar analyses.

Comments:

Figure1: the color scheme used in this figure makes it difficult to distinguish some of the genotypes, in particular the two ZI genotypes from the SU58xZI crosses. Please make these colors more distinct.

Figure 2 B: What is the overlap between the gene lists summarized in this table? How much sharing is there between the lists of DE genes between comparisons of the same type with respect to population or the F1s?

The microbiome composition is a nice addition and while I agree that removing Wolbachia seems reasonable in general I have concerns about how this affects the analysis. Is the difference in % Wolbachia significant between either the Sweden and Zambian populations or either parent and the F1 for the midgut?

Number of cell types in each tissue - this is mentioned briefly in the discussion, but I think a bit more context about the tissues would be appropriate here. Additional information about the number of cell types known in each of these tissues would help to give context to the findings in the paper.

The authors point out that there are differences in the inversion status between some of the lines, but do not address if there is any effect of the inversion in the regulatory variation. Is there, for example, an increase in trans effects within the In(2L)t region for comparisons involving SU26?

Reviewer #2: In this study, Glaser-Schmitt et al. examine variation in transcript abundance in three tissues (two functionally related ones and one somewhat less related) and in four genetic backgrounds of D. melanogaster to elucidate the evolution of regulatory architecture. To evaluate potential interactions between gene expression variation and microbiome communities, the authors also characterize bacterial composition in the two gut tissues. In both transcriptome and microbiome data, the authors detected pervasive genetic background- and tissue-specific effects, with the latter being typically stronger than the former. Analyses of allele specific expression (ASE) showed that genes with trans-effects are the most abundant regulatory class among genes with non-ambiguous regulatory variation. Not surprising, tissue specificity was higher for both cis- and trans-regulated genes than for genes with conserved expression across all genetic backgrounds. However, genes with trans-effects were more tissue-specific than genes with cis-effects. This observation suggests that variation in gene expression due to trans-regulatory elements can be spatially fine-tuned as well as (or even better than) that regulated by cis elements.

This is a well-written manuscript based on mRNAseq data analysis. The analyses are sound, and the figures of the manuscript are elaborated and successfully encapsulate the main points of the study in a clear and informative way, which is remarkable when we consider the large amount of new data reported. The results have a large descriptive component and yet they are interesting and exciting. This study fills a current gap in the knowledge of the field and will spark new research on the genetic basis of regulatory evolution. I don’t have any major issues, so I am recommending acceptance for publication in PLoS Genetics with minor revisions. I have just a few minor points and some editorial comments (see below).

1. The term dominance (h) divergence is used to describe variation in degree of dominance among genotypes (within the same tissue) but also among tissues (within the same genotype). It is not clear, however, how dominance divergence is calculated. Furthermore, the word divergence is usually reserved for differences between species but in this study, it also refers to differences among tissues so, in my view, it is a bit confusing. Perhaps the authors would consider alternative terms. “Dominance distance”? Along these lines, what is the difference between “Dominance” and “Magnitude Dominance” (Fig. 4)? Additional clarification is needed.

2. Following the previous point, the terms “gene expression divergence” are used to refer to differences in gene expression among tissues. However, “gene expression divergence” has been extensively used to describe differences in gene expression between species so it would be better to find an alternative. Maybe “gene expression distance?” Not an easy task.

3. This study focuses on gene expression in females. Is there sexually dimorphic gene expression in the 3 tissues examined in D. melanogaster? How much? I wonder about the relative contribution of genes with different degrees of sex-biased expression to the main findings of the study. One could potentially analyze male-, female- and nonsex-biased genes separately, and see to what extent conclusions hold. This may be relevant, considering that the different categories of sex-biased expression genes have been shown to have, for instance, different rates of protein evolution. The downside to this approach is perhaps the potential loss of statistical power. Just a thought for the authors to consider.

4. A technical comment. One of the observations of this study is that divergence for dominance is significantly higher than for gene expression (Bonferroni-corrected P < 10-14), which suggests that phenotypic dominance of expression is much less conserved among tissues and genotypes than expression itself (lines 275-276). What statistical test was used here? Did it consider that gene expression divergence is expected to be distributed between 0 and 1 while dominance divergence between -1 and 1? Please elaborate.

5. RNAseq data. Since poly-A selection was performed using total RNA, it should formally be mRNAseq data instead.

6. Figures are very informative. Fig. 1 & 7; I would suggest a more contrasting pattern of colors for increased clarity. Fig. 4 B; X-axis legend should be “Intersection class”, not “Interaction class”. Fig. 4C and D; the colors are helpful but I had difficulty seeing the asterisks of statistical significance so I would suggest increasing size a bit.

EDITORIAL COMMENTS

• Ln 103 “… single tissues, and/or highly diverged tissues….” replace with “… single tissues, and/or highly functionally diverged tissues….”

• Ln 132 “… From ASE data, we found that both cis and trans effects were more tissue-specific than genes with no differential expression…” replace with “… From ASE data, we found that genes with both cis and trans effects were more tissue-specific than genes with no differential expression… “

• Ln 151 “ … some overlap between SU58, ZI418 and their F1 hybrid and well as …” replace with “ … some overlap between SU58, ZI418 and their F1 hybrids, and as well as …”

• Ln 249 “In order to compare the magnitude of dominance regardless of which allele was dominant, we calculated h such that values between 0 and 1 or 0 and -1 represent varying degrees of additivity and dominance with values closer to 1 or -1 being more dominant and -1 representing complete dominance of the Swedish background and 1 representing complete dominance of the Zambian background, while values outside this range represent cases of overdominance of the respective background (see Methods for more details).” replace with “In order to compare the magnitude of dominance regardless of which allele was dominant, we calculated h such that values between 0 and 1 or 0 and -1 represent varying degrees of additivity and dominance, with values closer to -1 representing complete dominance of the Swedish background and 1 representing complete dominance of the Zambian background, while values outside this range represent cases of overdominance of the respective background (see Methods for more details).”

• Ln 351 “… and this was only significant after multiple test correction in the midgut…” replace with “… and this was significant after multiple test correction only in the midgut…”

• Ln 753 “ … the ASE analysis section above in order balance statistical power…” replace with “ … the ASE analysis section above in order to balance statistical power…”

**Have all data underlying the figures and results presented in the manuscript been provided?**

Reviewer #1: Yes

Reviewer #2: Yes

PLOS authors have the option to publish the peer review history of their article (what does this mean?). If published, this will include your full peer review and any attached files.

Reviewer #1: No

Reviewer #2: No

---

## [Editor Report · Decision Letter 1]

30 Jul 2024

Dear Dr. Glaser-Schmitt,

We are pleased to inform you that your manuscript entitled "Pervasive tissue-, genetic background-, and allele-specific gene expression effects in *Drosophila melanogaster*" has been editorially accepted for publication in PLOS Genetics. Congratulations!

Yours sincerely,

Trudy F.C. Mackay

Consulting Editor - PLoS Genetics

PLOS Genetics

Scott Williams

Section Editor

PLOS Genetics

**Data Deposition**

http://datadryad.org/submit?journalID=pgenetics&manu=PGENETICS-D-24-00414R1

**Press Queries**

---

## [Editor Report · Acceptance letter]

17 Aug 2024

PGENETICS-D-24-00414R1 

Pervasive tissue-, genetic background-, and allele-specific gene expression effects in *Drosophila melanogaster*

Dear Dr Glaser-Schmitt, 

We are pleased to inform you that your manuscript entitled "Pervasive tissue-, genetic background-, and allele-specific gene expression effects in *Drosophila melanogaster*" has been formally accepted for publication in PLOS Genetics! Your manuscript is now with our production department and you will be notified of the publication date in due course.

With kind regards,

Zsofia Freund

PLOS Genetics

On behalf of:
